

# Loop equations for generalised eigenvalue models

Edoardo Vescovi[1][⋆] and Konstantin Zarembo[1,2][†]

**1** Nordita, KTH Royal Institute of Technology and Stockholm University,
Hannes Alfvéns väg 12, 106 91 Stockholm, Sweden
**2** Niels Bohr Institute, Copenhagen University,
Blegdamsvej 17, 2100 Copenhagen, Denmark

⋆ edoardo.vescovi@su.se , † zarembo@nordita.org

## Abstract

We derive the loop equation for the 1-matrix model with generic difference-type measure for eigenvalues and develop a recursive algebraic framework for solving it to an arbitrary order in the coupling constant in and beyond the planar approximation. The planar limit is solved exactly for a one-parametric family of models and in the general case at strong coupling. The Wilson loop in the $\mathcal{N} = 2^*$ super-Yang-Mills theory and the Hoppe model are used to illustrate our methods.

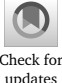

# 1   Introduction

The loop equations were originally proposed for the Yang-Mills theory [1, 2] in view of its potential reformulatation as a theory of strings [3]. Their zero-dimensional counterpart [2, 4], in the same vein, gives a non-perturbative definition of quantum gravity in two dimensions [5–7]. The starting point then is a zero-dimensional quantum field theory, a matrix model. The loop equations are the Schwinger-Dyson equations for the Wilson loop expectation values that follow from reparametrisation invariance of matrix averages. Their perturbative (in $1/N^2$) solution generates the topological expansion of correlation functions with interpretation in terms of discretised random surfaces [3].

In the planar limit of strictly infinite $N$, matrix models can be solved by a variety of techniques such as the saddle-point method [8] or orthogonal polynomials [9,10]. The added value of the loop equations is an ease in going beyond the planar approximation [11]. A powerful recursive procedure developed in [12] can be systematically extended to any desired order in $1/N^2$, and lies at the heart of the topological recursion formulated entirely in geometrical terms [13].

In this paper we study loop equations for random eigenvalue ensembles

$$Z = \int\limits_{-\infty}^{+\infty} \prod_{i=1}^{N} da_i \prod_{i<j} \mu(a_i - a_j)\, e^{-N\sum\limits_i V(a_i)}, \tag{1}$$

with $\mu(a)$ and $V(a)$ subject to the obvious convergence conditions but otherwise arbitrary. This class of models is a particular case of more general integrals with bi-local measure [14, 15] with the measure function of difference type.

Many such models have been studied in the past. The standard Vandermonde measure $\mu(a) = a^2$ corresponds to Hermitian random matrices. A one-parameter deformation thereof, $\mu(a) = a^{2\beta}$, is known as the $\beta$-ensemble. The topological recursion for it was formulated

in [16,17] and further studied in [18–21]. Other solvable cases include the Hoppe model $\mu(a) = a^2/(a^2 + m^2)$ [22,23], and the Chern-Simons matrix model with $\mu(a) = \sinh a$ [24–26].

The topological recursion can in fact be formulated for an arbitrary bi-local measure [14], with the genus-zero expectation values considered as an input. It is unlikely that the genus-zero problem is solvable for arbitrary $\mu(a)$ and $V(a)$, and our goal here is more modest – to study the loop equations in the regimes where a well-defined approximation scheme can be devised, namely at weak and at strong coupling.

The eigenvalue integrals with the difference-type measure arise in many contexts: quantum D-branes [23], supersymmetric Wilson loops [27,28], supersymmetric localisation [29], Chern-Simons theory [24,25], knot invariants [30,31] and many more. To give one example, $\mathcal{N} = 2$ supersymmetric gauge theories on $S^4$ are described by eigenvalue integrals with the Gaussian potential and a fairly complicated measure: for the mass deformation of the $\mathcal{N} = 4$ super-Yang-Mills known as the $\mathcal{N} = 2^*$ theory, the measure is expressed through a combination of the Meijer G-functions [29]. The latter model has been studied by a variety of approximate techniques [32–38] but has never been solved in full generality and will be a prime example to illustrate our methods. Localisation integrals in superconformal gauge theories [39–46] constitute another set of matrix models where a complicated measure precludes the use of conventional techniques, and we believe that our methods will be useful in this context as well.

Section 2.1 covers the basics to the inverse Laplace transforms of $n$-point resolvents, also called $n$-point Wilson loop operators $\mathbb{W}(x_1, x_2, \ldots, x_n)$. Section 2.2 derives the (non-linear, integro-differential) loop equation for arbitrary $N$ and polynomial potential. It closes on the single function $\mathbb{W}(x)$ in the planar limit and depends also on $\mathbb{W}(x, y)$ otherwise. In preparation for large-$\lambda$ models, section 2.3 converts the equation for a one-parameter deformation of the Vandermonde measure into an exactly-solvable algebraic problem. Section 3.1 operates a change of variables to eliminate the linear term in the potential. This is instrumental in the structure of the perturbative solution in section 3.2 and in the algebraic equations for the series coefficients in section 3.3. We collect and test our results for $\mathbb{W}(x)$ to high orders against exact formulas in various models in section 4. The key observation that makes contact with physics is the fact that certain (matrix-model) Wilson loop operators evaluated at $x = 2\pi$ calculate the expectation value of the (gauge-theory) 1/2-BPS circular Wilson loop in $\mathcal{N} = 4$ and $\mathcal{N} = 2^*$ SYM theories. In particular section 4.3.2 pays great attention to the latter: an analysis of the perturbative convergence and a Padé resummation to extrapolate to $\lambda \gg 1$, which build on our power-series expansion of the planar Wilson loop to the order $\lambda^{35}$. For small mass parameter, the ratio test signals a singularity at $\lambda = -\pi^2$ which also affects other observables in $\mathcal{N} = 2$ theories. A finite value of the mass slows down the convergence rate and falls short of detecting a mass-dependent convergence radius. The large-mass/flat-space limit is incompatible with $\lambda \ll 1$ but formally organises the power series in agreement with the renormalisation of $\mathcal{N} = 2$ theories. Section 5 is about a systematic method to generate the expansion of $\mathbb{W}(x)$ in strongly-coupled models with a general deformation of the Vandermonde measure. We distinguish two regimes: when $x$ is smaller than the characteristic eigenvalue scale in sections 5.1 and 5.2.1, the leading approximation tracks back to section 2.3, whereas the opposite limit in sections 5.2 and 5.2.1 is brought to a form reminiscent of a Wiener-Hopf problem. We intersperse remarks for Gaussian models in this part and present a thorough application to the strongly-coupled Hoppe model in section 6. Section 7 outlines future advancements and the appendices collate technical proofs. The Mathematica notebooks attached to the publication automatise the algorithm of section 3.3 and generate the results of section 4.

## 2  Loop equation

### 2.1  Matrix model

The Wilson loop expectation is an exponential average:

$$\mathbb{W}(x) = \frac{1}{N}\left\langle \operatorname{tr} e^{xA} \right\rangle, \tag{2}$$

in the ensemble defined by the partition function (1). The two-point connected correlator is defined as

$$\mathbb{W}(x,y) = \left\langle \operatorname{tr} e^{xA} \operatorname{tr} e^{yA} \right\rangle - \left\langle \operatorname{tr} e^{xA} \right\rangle \left\langle \operatorname{tr} e^{yA} \right\rangle. \tag{3}$$

The averages are computed as multiple integrals

$$\langle f(a_1, \ldots, a_N) \rangle = \frac{1}{Z} \int_{-\infty}^{+\infty} \prod_i da_i \prod_{i<j} \mu(a_i - a_j) f(a_1, \ldots, a_N) e^{-N \sum_i V(a_i)}, \tag{4}$$

over the eigenvalues $a_i \in \mathbb{R}$ ($i, j = 1, 2, \ldots, N$) of the $N \times N$ hermitian matrix $A$. The integral measure $\mu(a)$ is a smooth function with $\mu(a) = \mu(-a) \geqslant 0$. The matrix model (4) encompasses the $\beta$-ensemble ($\mu(a) = a^{2\beta}$), including the Hermitian matrix model with unitary symmetry $U(N)$ (with Vandermonde measure, $\beta = 1$), the real symmetric matrices with orthogonal symmetry $SO(N)$ ($\beta = 1/2$) and the real quaternionic matrices with symplectic symmetry $Sp(N)$ ($\beta = 2$). We will later impose mild assumptions on the measure to achieve results in different regimes. The probability measure is set by the potential $V(a)$, given by a formal sum

$$V(a) = \frac{8\pi^2}{\lambda} \sum_{n=0}^{\infty} T_n a^n. \tag{5}$$

We will need all "times" $T_n$ to generate loop averages by differentiation, upon which we assume the potential settles to some fixed polynomial with only finite number of terms present. We factor out the expansion parameter $\lambda$, which possibly takes small or large values, from the "coupling constants" $T_n$, which are finite and include odd $n$, thus allowing non-symmetric potentials. When eigenvalues are seen as the coordinates of $N$ particles along a line, the $i$-th particle is subject to the potential $V(a_i)$ and each pair repels with energy $-\log \mu(a_i - a_j)$. Large values of $\lambda$ spread particles apart and away from the potential minima in the equilibrium configuration. For a more detailed account of random matrix models we refer to reviews [47–49].

The loop correlators $\mathbb{W}(x)$ and $\mathbb{W}(x,y)$ are the first two representatives of the family

$$\mathbb{W}(x_1, \ldots, x_n) = N^{n-2} \left\langle \operatorname{tr} e^{x_1 A} \ldots \operatorname{tr} e^{x_n A} \right\rangle_{\mathrm{c}}, \tag{6}$$

where the subscript "c" pertains to the connected part [20]. A more commonly used set of observables are correlators of trace resolvents, related to the exponential correlators by the inverse Laplace transform. Namely,

$$\omega(x) = \frac{1}{N} \left\langle \operatorname{tr} \frac{1}{x-A} \right\rangle. \tag{7}$$

The multi-point correlators of resolvents are generating functions of the basic monomial correlators:

$$\omega(x_1, \ldots, x_n) = N^{n-2} \left\langle \operatorname{tr} \frac{1}{x_1 - A} \ldots \operatorname{tr} \frac{1}{x_n - A} \right\rangle_{\mathrm{c}} \tag{8}$$

$$= N^{n-2} \sum_{k_1, \ldots, k_n = 0}^{\infty} x_1^{-k_1 - 1} \ldots, x_n^{-k_n - 1} \left\langle \operatorname{tr} A^{k_1} \ldots \operatorname{tr} A^{k_n} \right\rangle_{\mathrm{c}}.$$

Like the partition function, $\mathbb{W}(x)$ and $\mathbb{W}(x,y)$ are formal power series in the potential coefficients by the expansion of (5) in the vicinity of a minimum. They are generating functions similar to (7) and (8):

$$\mathbb{W}(x) = \frac{1}{N} \sum_{n=0}^{\infty} \frac{x^n}{n!} \langle \mathrm{tr} A^n \rangle , \tag{9}$$

$$\mathbb{W}(x,y) = \sum_{n,m=0}^{\infty} \frac{x^n y^m}{n!m!} (\langle \mathrm{tr} A^n \, \mathrm{tr} A^m \rangle - \langle \mathrm{tr} A^n \rangle \langle \mathrm{tr} A^m \rangle), \tag{10}$$

and in that they have a $1/N$-expansion, called topological expansion [7]. The product of $n$ traces enumerates discrete open surfaces with $n$ holes, whose Euler characteristics control the power of $N$, and the variables $x_1, \ldots, x_n$ are fugacities for the lengths of the $n$ boundaries [8,50–52]. The expansion in the couplings is associated with ribbon-graph diagrams [3].

The genus expansion begins at $N^0$ in our normalisation [16,21] and odd powers of $N^{-1}$ are absent when the measure scales as $\mu(a) \sim Ca^2$ for small eigenvalue separation (the constant $C$ is not important for this conclusion), in accordance with the observation that only the Hermitian model has this property among the $\beta$-ensembles. Such measure defines a deformation of the Vandermonde determinant and brings us to coin the name *generalised eigenvalue model* for (4).

Although the focus of section 2.2 is on an equation for $\mathbb{W}(x)$, this has to be fed with $\mathbb{W}(x,y)$, save for the strict large-$N$ limit. The loop-insertion method [53, 54] generates all loop correlators by consecutive differentiation:

$$\mathbb{W}(x_1, \ldots, x_n) = N^{-2} \frac{\delta}{\delta V(x_1)} \cdots \frac{\delta}{\delta V(x_n)} \log Z \tag{11}$$

$$= \frac{\delta}{\delta V(x_1)} \cdots \frac{\delta}{\delta V(x_{n-1})} \mathbb{W}(x_n),$$

where the loop-insertion operator is defined as[1]

$$\frac{\delta}{\delta V(z)} = -\frac{\lambda}{8\pi^2} \sum_{n=0}^{\infty} \frac{z^n}{n!} \frac{\partial}{\partial T_n} . \tag{12}$$

The simplicity of the recursion comes at the cost of solving the loop equation for an infinite-degree potential (5) before choosing a specific polynomial.

## 2.2 Loop equation

We derive the loop equation (27) below: a non-linear integro-differential equation for $\mathbb{W}(x)$, which depends on an infinite-degree potential (5) and a measure function $\mu$. The latter is subject to $\gamma(0) = 2$, where we find convenient to define the even function[2]

$$\gamma(a) = a \frac{d}{da} \log \mu(a). \tag{13}$$

As noted below (10), this puts a mild constraint on the topological expansion. Far from being necessary, it is motivated by our interest in supersymmetric theories.

---

[1] It is the inverse Laplace transform of the operator $-\frac{\lambda}{8\pi^2} \sum_{n=0}^{\infty} z^{-n-1} \partial/\partial T_n$, which similarly acts upon (7) to generate the tower of multi-point (or multi-loop, in matrix-model terminology) resolvents (8).

[2] The condition $\gamma(0) = 2$ is necessary to have a regular large-$N$ expansion with only even powers of $1/N$ appearing. This technical assumption, that immediately excludes all $\beta$-ensembles, becomes important once we take into account $1/N$ corrections. All our result at leading planar approximation are independent of it and are valid for models with $\gamma(0) \neq 2$ as well.

We will use also another function

$$R(a) = \frac{1}{2} \frac{d}{da} \log \mu(a) = \frac{\gamma(a)}{2a}. \tag{14}$$

We assume that these two functions admit regular Fourier representation:

$$\gamma(a) = \int\limits_{-\infty}^{+\infty} \frac{d\omega}{2\pi} e^{-i\omega a} \hat{\gamma}(\omega), \qquad R(a) = \int\limits_{-\infty}^{+\infty} \frac{d\omega}{2\pi i} e^{-i\omega a} \hat{R}(\omega). \tag{15}$$

The function $\hat{R}$ is anti-symmetric:

$$\hat{R}(-\omega) = -\hat{R}(\omega), \tag{16}$$

and real, with the above definition.

The resolvent obeys loop equations, also called Pastur equations in mathematics and Schwinger-Dyson equations in the physics literature. The loop equations are self-consistency conditions based off the invariance of (4) under an infinitesimal local change of variables or integration by parts. We adapt the latter derivation [6, 20] (usually done for the resolvent) to the exponential average. The shift of focus draws inspiration from the loop dynamics in a QCD-like random matrix model [2].

We insert an exponential in the form of (2) into the average

$$\frac{1}{Z} \int\limits_{-\infty}^{+\infty} \prod_i da_i \sum_p \frac{\partial}{\partial a_p} \left[ \prod_{i<j} \mu(a_i - a_j) e^{x a_p} e^{-N \sum_i V(a_i)} \right] = 0. \tag{17}$$

The chain rule delivers three terms. The derivative acting on the potential yields

$$-N \left\langle \sum_p V'(a_p) e^{x a_p} \right\rangle = -N^2 V'\left(\frac{d}{dx}\right) \mathbb{W}(x). \tag{18}$$

We swap the average and the operator

$$V'\left(\frac{d}{dx}\right) = \frac{8\pi^2}{\lambda} \sum_{n=1}^{\infty} n T_n \left(\frac{d}{dx}\right)^{n-1}. \tag{19}$$

The derivative acts on $\mu$ and the exponential as

$$\left\langle x \sum_p e^{x a_p} \right\rangle + \left\langle \sum_{i \neq p} e^{x a_p} \frac{\mu'(a_p - a_i)}{\mu(a_p - a_i)} \right\rangle. \tag{20}$$

We recall (2), use $\gamma(a) = \gamma(-a)$ to symmetrise in $i \leftrightarrow p$

$$N x \mathbb{W}(x) + \frac{1}{2} \left\langle \sum_{i \neq p} \frac{e^{x a_p} - e^{x a_i}}{a_p - a_i} \gamma(a_p - a_i) \right\rangle, \tag{21}$$

complete the sum with the terms with $i = p$

$$\left(1 - \frac{\gamma(0)}{2}\right) N x \mathbb{W}(x) + \frac{1}{2} \left\langle \sum_{i,p} \frac{e^{x a_p} - e^{x a_i}}{a_p - a_i} \gamma(a_p - a_i) \right\rangle, \tag{22}$$

and notice that the assumption above (13) removes the linear dependence on $N$. The next step is to link up to the functionals (3), noting the double sum, and possibly (2). For that one has to write the expression as a combination of $e^{x a_i}$. The way to achieve this is not unique.[3] While the numerator is already in this form, we operate the identity

$$\frac{1}{a_p - a_i} = \int_0^x ds\, e^{s(a_p - a_i)},\qquad(23)$$

on the denominator and trade the measure factor for its Fourier transform

$$\gamma(a) = \int_{-\infty}^{+\infty} \frac{d\omega}{2\pi}\, e^{-i\omega a}\hat{\gamma}(\omega),\qquad \hat{\gamma}(\omega) = \int_{-\infty}^{+\infty} da\, e^{i\omega a}\gamma(a).\qquad(24)$$

This brings (22) to the desired form

$$\frac{1}{2}\sum_{i,p}\int_{-\infty}^{+\infty}\frac{d\omega}{2\pi}\hat{\gamma}(\omega)\int_0^x ds\,\left\langle e^{s a_p + (x-s)a_i}e^{i\omega(a_p - a_i)}\right\rangle,\qquad(25)$$

at the cost of introducing two auxiliary integrations. The comparison with (3) yields explicitly the Wilson loops

$$\frac{1}{2}\int_{-\infty}^{+\infty}\frac{d\omega}{2\pi}\hat{\gamma}(\omega)\int_0^x ds\,\left[\mathbb{W}(s-i\omega, x-s+i\omega) + N^2\mathbb{W}(s-i\omega)\mathbb{W}(x-s+i\omega)\right].\qquad(26)$$

The last step puts together (18) and (26) and yields the loop equation[4]

$$V'\!\left(\frac{d}{dx}\right)\mathbb{W}(x) = \frac{1}{2}\int_{-\infty}^{+\infty}\frac{d\omega}{2\pi}\hat{\gamma}(\omega)\int_0^x ds\,\left[\mathbb{W}(s-i\omega)\mathbb{W}(x-s+i\omega) + \frac{\mathbb{W}(s-i\omega, x-s+i\omega)}{N^2}\right].$$

$$(27)$$

The formula deserves a number of comments.

First, the equation displays a direct dependence on the model (4): the parameters (number of eigenvalues $N$, or rank of gauge group $U(N)$, and coupling $\lambda$), the integral measure (via the Fourier transform (24) of its logarithmic derivative (13)) and potential (via the infinitely-many coefficients $\{T_n\}_{n=1,2,...}$ of (19)). The two-point operator $\mathbb{W}(x, y)$ is linear in the solution through one loop insertion (11) and (12):

$$\mathbb{W}(x, y) = \frac{\delta\mathbb{W}(y)}{\delta V(x)}.\qquad(28)$$

Second, the equation is to solve for $\mathbb{W}(x)$ on the real line. The normalisation of (2) sets the initial condition $\mathbb{W}(0) = 1$. In section 4 the matrix-model quantity $\mathbb{W}(2\pi)$ maps to physical observables (expectation values of gauge-theory Wilson loops), hence one can restrict to the segment $x \in [0, 2\pi]$. For this to be possible, our derivation prefers (23) among equivalent representations with domain larger than $s \in [0, x]$.[5] The imaginary shifts $\pm i\omega$ require the analytical continuation to the strip $\mathrm{Re}(x) \in [0, 2\pi]$ in the complex plane.

---

[3]Any integral with exponential weight, like Fourier and Laplace transforms of a kernel, fits the criteria. We present a simpler version of the loop equation later in this section exploiting this freedom.

[4]This is the first one in a chain of loop equations that intertwines higher-point Wilson loops.

[5]It is hard to devise representations with $s \in [0, \infty]$ that converge for all orderings $a_p - a_i \lessgtr 0$.

Third, the equation is non-linear (due to the first term in the integrand) and of a specific integro-differential type (thus backed up by little literature). The imaginary shifts prevent the interpretation of the $s$-integral as a convolution, hence precluding a solution in Laplace transform. An exception is the subject of section 4.1: the Vandermonde measure $\mu(a) = a^2$ maps to a Dirac delta $\hat{\gamma}(\omega) = 4\pi\delta(\omega)$, which localises the integrand on vanishing shifts.

Fourth, there are two ways to approach the equation perturbatively. At infinite $N$, the operators $\mathbb{W}(x)$ and $\mathbb{W}(x, y)$ are of order $N^0$ and the latter drops out of (27). At large $N$, they expand in series of $N^{-2}$. The suppression factor $N^{-2}$ in (27) helps solving order by order in an intertwined manner. The knowledge of $\mathbb{W}(x)$ up to $N^{-2n}$ calculates $\mathbb{W}(x, y)$ up to $N^{-2n}$ by loop insertion, which is necessary in turn to write the equation at the order $N^{-2(n+1)}$ and find the next term $N^{-2(n+1)}$ of $\mathbb{W}(x)$.

The same argument applies to the expansion in small $\lambda$: the left-hand side contains powers up to $\lambda^{\ell-1}$ and the right-hand side $\lambda^\ell$ at the iteration $\ell = 0, 1, \ldots$, hence one iteratively reads off the order $\lambda^\ell$ from the former. The argument relies on the linearity of the left-hand side to easily adjust the next unknown term.

The same reasoning does not go through at large coupling because the right-hand side is not linear. Once a truncated solution is plugged into the left-hand side, one would have to disentangle the next order from the integral equation on the right-hand side. The obstruction is intrinsic to the loop equation: it is not washed away by large $N$ nor by a choice of model. Section 5 revisits the equation with a method similar to the Wiener-Hopf decomposition.

Another form of the loop equation arises upon Fourier transforming $R(a)$ instead of $\gamma(a)$ and is best formulated in terms of the oscillating loop average:

$$W(\kappa) = \frac{1}{N}\left\langle \sum_k e^{i\kappa a_k} \right\rangle, \tag{29}$$

related to the Wilson loop in (2) by an analytic continuation: $\mathbb{W}(x) = W(-ix)$. Repeating the steps following (17) with the insertion of $e^{i\kappa a_p}$ and using the anti-symmetry of $\hat{R}(\omega)$ we find:

$$\frac{i}{2} V'\left(-i\frac{\partial}{\partial \kappa}\right) W(\kappa) = \int_{-\infty}^{+\infty} \frac{d\omega}{2\pi} \hat{R}(\omega) W(\kappa - \omega) W(\omega). \tag{30}$$

This manifestly real form of the equation (for an even potential) is particularly beneficial for the strong-coupling analysis in section 5. The non-planar corrections, dropped here for simplicity, are easily recovered by replacing $WW$ with $WW + N^{-2}\delta W/\delta V$.

This Fourier-space form of the loop equation is slightly unconventional, it has no obvious symmetries, and even for the Vandermonde measure with $\hat{R}(\omega) = -\pi\,\mathrm{sign}\,\omega$ contains a non-linear integral transform that does not reduce to a simple convolution. In contrast, the kernel in (27) for the Vandermonde measure becomes the delta-function $\hat{\gamma}(\omega) = 4\pi\delta(\omega)$ and the loop equation assumes the standard convolution form. In the appendix A we show that the alternative form of the loop equation (30) is equivalent to the standard saddle-point conditions of the original eigenvalue model.

Below we identify a class of models, generalising the usual matrix model with the Vandermonde measure, for which (30) can be solved exactly.

## 2.3 The $\nu$-model

We start by considering an eigenvalue model with a polynomial potential and the kernel

$$\hat{R}(\omega) = -\pi\,\mathrm{sign}\,\omega\,|\omega|^{2\nu-2}. \tag{31}$$

The conventional Vandermonde measure corresponds to $\nu = 1$. Fourier transforming back to the $a$-representation gives a rather baroque-looking measure. Nonetheless the loop equation is solvable in this case. This solution will be an integral part of the strong-coupling analysis for generic $\hat{R}(\omega)$. We thus spend some time discussing this type of models which we collectively call the *$\nu$-model*.

The loop equation with the above kernel can be written as

$$-i\,V'\left(-i\frac{\partial}{\partial\kappa}\right)W(\kappa) = \int_0^\infty d\omega\,\omega^{2\nu-2}W(\omega)\big(W(\kappa-\omega)-W(\kappa+\omega)\big)\,. \tag{32}$$

The exact solution of this equation can be found by exploiting the following mathematical fact.

Consider a function $f(\kappa)$ whose Fourier transform has a compact support:

$$f(\kappa) = \int_{-1}^{1} dt\,e^{it\kappa}\hat{f}(t)\,. \tag{33}$$

Then the following operator identity holds true for any integer $n$:

$$\int_0^\infty d\omega\,\omega^{2\nu-2}\frac{J_{\nu+n}(\omega)}{\omega^{\nu+n}}\big(f(\kappa-\omega)-f(\kappa+\omega)\big) = -i\mathcal{D}_n^{(\nu)}f(\kappa)\,, \tag{34}$$

where $\mathcal{D}_n^{(\nu)} \equiv \mathcal{D}_n^{(\nu)}(-i\partial/\partial\kappa)$ is a differential operator of finite degree. For $n$ negative it vanishes identically:

$$\mathcal{D}_n^{(\nu)} = 0 \qquad (n < 0)\,, \tag{35}$$

and for $n$ positive or zero $\mathcal{D}_n^{(\nu)}$ is a differential operator of degree $2n+1$ expressed through the Gegenbauer polynomials:

$$\mathcal{D}_n^{(\nu)}(a) = (-1)^n 2^{\nu-n-1}\Gamma(\nu-n-1)C_{2n+1}^{(\nu-n-1)}(a)\,. \tag{36}$$

The proof is surprisingly simple, and goes by direct calculation the details of which are given in the appendix B.

We list here the first few $\mathcal{D}_n^{(\nu)}$ polynomials:

$$\mathcal{D}_0^{(\nu)}(a) = 2^\nu\Gamma(\nu)a\,, \qquad \mathcal{D}_1^{(\nu)}(a) = \frac{2^\nu\Gamma(\nu)}{6}\left(3a-2\nu a^3\right)\,,$$

$$\mathcal{D}_2^{(\nu)}(a) = \frac{2^\nu\Gamma(\nu)}{120}\left[15a-20\nu a^3+4\nu(\nu+1)a^5\right]\,. \tag{37}$$

The general definition and key properties of the Gegenbauer polynomials $C_k^{(\lambda)}$ can be found in [55].

The integral transform in (34) suggests to choose $J_{\nu+n}(\mu\kappa)/(\mu\kappa)^{\nu+n}$ as the basic building blocks of the Wilson loop and to seek the solution of the loop equation as linear combination of these functions. We thus take an Ansatz of the form:

$$W(\kappa) = \sum_{n\geqslant 0} A_n\frac{J_{\nu+n}(\mu\kappa)}{(\mu\kappa)^{\nu+n}}\,. \tag{38}$$

The rescaling factor $\mu$ is necessary for imposing the normalisation condition $W(0) = 1$. Physically, $\mu$ corresponds to the endpoint of the eigenvalue distribution as we detail below.

The rationale to take this Ansatz is twofold. First, all the functions in the basis have a Fourier image with a compact support, as follows from the integral representation of the Bessel function:

$$\frac{J_{\nu+n}(\kappa)}{\kappa^{\nu+n}} = \frac{1}{2^{\nu+n}\sqrt{\pi}\,\Gamma\left(\nu+n+\frac{1}{2}\right)}\int\limits_{-1}^{1} dt\, e^{it\kappa}\left(1-t^2\right)^{\nu+n-\frac{1}{2}}.\tag{39}$$

The Ansatz therefore satisfies the condition (33). At the same time, regarding $W(\omega)$ in (32) as the kernel of an integral operator we can apply the master identity (34) valid for all individual Bessel kernels and a function with finite support.

The operator identity (34) transforms (32) into a linear differential equation:

$$V'\left(-i\frac{\partial}{\partial\kappa}\right)W(\kappa) = \mu^{1-2\nu}\sum_n A_n \mathcal{D}_n^{(\nu)}\left(-\frac{i}{\mu}\frac{\partial}{\partial\kappa}\right)W(\kappa).\tag{40}$$

Both sides are differential polynomials acting on one and the same function. It just remains to match the coefficients:

$$V'(\mu a) = \mu^{1-2\nu}\sum_n A_n \mathcal{D}_n^{(\nu)}(a).\tag{41}$$

Solving the loop equation boils down to a simple algebraic problem: given a polynomial $V'(\mu a)$, find its expansion coefficients in the basis (36). In each particular case the problem is easily solved by hand, just by matching the coefficients one-by-one, but it is also possible to derive the general solution by exploiting pseudo-orthogonality of the Gegenbauer polynomials. We first illustrate the matching procedure and then derive the general solution using orthogonality.

Consider the Gaussian potential:

$$V(a) = \frac{a^2}{2g^2}.\tag{42}$$

The left-hand side of (41) is then linear in $a$ and the first polynomial in (37) will suffice. Matching the coefficient we get:

$$A_0 = \frac{\mu^{2\nu}}{2^\nu \Gamma(\nu)g^2},\tag{43}$$

which yields for the Wilson loop:

$$W(\kappa) = \frac{1}{\Gamma(\nu)g^2}\left(\frac{\mu}{2\kappa}\right)^\nu J_\nu(\mu\kappa).\tag{44}$$

This still contains an unknown parameter $\mu$, and to fix it we need to impose the normalisation condition $W(0) = 1$.

Taking into account that

$$\lim_{u\to 0}\frac{J_\nu(u)}{u^\nu} = \frac{1}{2^\nu \Gamma(\nu+1)},$$

we get:

$$\left(\frac{\mu}{2}\right)^{2\nu} = \Gamma(\nu)\Gamma(\nu+1)g^2.\tag{45}$$

The properly normalised solution thus reads

$$W(\kappa) = \Gamma(\nu+1)\frac{J_\nu(\mu\kappa)}{(\mu\kappa/2)^\nu}.\tag{46}$$

The last two equations solve the $\nu$-model with the Gaussian potential.

When $\nu = 1$ the relation between $\mu$ and $g$ is linear: $\mu = 2g$. Upon continuation to complex $\kappa = -ix$ we recover the well-known expression for the Wilson loop in the Gaussian matrix model:

$$\mathbb{W}(x) = \frac{I_1(2gx)}{gx}. \tag{47}$$

We revisit this result in section 4.3.1 from a slightly different perspective.

Returning to the generic case, an expansion of a given polynomial $V'$ in the basis (36) can be derived from pseudo-orthogonality of the Gegenbauer polynomials. The polynomial basis (36) does not form an orthonormal set with respect to any measure, for the reasons explained in the appendix B. This can be also verified directly by checking that $\mathcal{D}_n^{(\nu)}$ do not satisfy trilinear relations characteristic of orthogonal polynomials, as can be seen by examining the first few cases. The closest counterpart of functional orthogonality is derived in the appendix B, and involves a differential operator inside the integral:

$$\int_{-1}^{1} da\,(1-a^2)^{\nu-n-\frac{3}{2}}\mathcal{D}_n^{(\nu)}(a)\left(a\,\frac{\partial}{\partial a}+2\nu-1\right)\mathcal{D}_m^{(\nu)}(a) = \frac{4\pi\Gamma(2\nu-1)}{(2n+1)!}\,\delta_{nm}. \tag{48}$$

Applying the differential operator $a\partial/\partial a + 2\nu - 1$ to both sides of (41) and integrating with the prescribed weight, we get:

$$A_n = \frac{\mu^{2\nu-1}(2n+1)!}{4\pi\Gamma(2\nu-1)}\int_{-1}^{1} da\,(1-a^2)^{\nu-n-\frac{3}{2}}\mathcal{D}_n^{(\nu)}(a)\left(a\,\frac{\partial}{\partial a}+2\nu-1\right)V'(\mu a). \tag{49}$$

Together with (38) this constitutes the full solution of the $\nu$-model with any potential.

The real-valued Wilson loop is obtained by analytic continuation to $\kappa = -ix$ which transforms the entries in (38) into the modified Bessel functions:

$$\mathbb{W}(x) = \sum_{n\geqslant 0} A_n\,\frac{I_{\nu+n}(\mu x)}{(\mu x)^{\nu+n}}. \tag{50}$$

The width of the eigenvalue distribution $\mu$ is fixed by imposing the normalisation condition $\mathbb{W}(0) = 1$:

$$\sum_n \frac{A_n}{2^{\nu+n}\Gamma(\nu+n+1)} = 1. \tag{51}$$

We can also find the eigenvalue density directly. The density is formally defined as a Fourier transform of the Wilson loop:

$$W(\kappa) = \int_{-\mu}^{\mu} da\, e^{i\kappa a}\rho(a). \tag{52}$$

Using the integral representation (39) in (38) we arrive at

$$\rho(a) = \frac{1}{\sqrt{\pi}}\sum_n \frac{A_n(\mu^2-a^2)^{\nu+n-\frac{1}{2}}}{(2\mu^2)^{\nu+n}\Gamma\left(\nu+n+\frac{1}{2}\right)}. \tag{53}$$

We have checked that for $\nu = 1$ this formula, together with the equation for $A_n$, boils down to the well-known solution of the conventional Vandermonde matrix model:

$$\rho(a) = \int_{-\mu}^{\mu} \frac{db}{2\pi^2}\,\frac{V'(b)}{b-a}\sqrt{\frac{\mu^2-a^2}{\mu^2-b^2}}. \tag{54}$$

The eigenvalue density must be positive on the whole interval $(-\mu, \mu)$ which imposes certain constraints on the coefficients $A_n$.[6] The simplest of those is the positivity of the density near the endpoints which requires

$$A_0 > 0. \tag{55}$$

Indeed, $A_0$ multiplies the smallest power of $(\mu^2 - a^2)$ in (53). The other terms (those with $n > 0$) are necessarily more suppressed for $a$ very close to $\mu$. If $A_0 < 0$ the density is destined to becomes negative at some $a = \mu - \epsilon$ with small $\epsilon$. Therefore, the model undergoes a transition into a multi-cut phase as soon as $A_0$ reaches zero. The multi-cut solutions are not described by our Ansatz.[7] It would be very interesting to explore the multi-cut phases of the $\nu$-model by extending our method to this case, or by some other technique.

For generic $V(a)$ and $\hat{R}(\omega)$ the non-linear integro-differential equation (30) cannot be solved in a closed form, but the weak and strong coupling expansions can be worked out systematically under very general assumptions. This is what we do in the rest of the paper.

## 3 Weak coupling: Solution

### 3.1 Shift of the matrix variables

In preparation for the perturbative analysis, we eliminate the linear term $n = 1$ in (5) and change the loop equation accordingly. In quantum models linearities conflict with the interpretation of $\{a_i\}_{i=1,2,\dots,N}$ as good perturbative degrees of freedom. The shift $\tilde{a}_i = a_i - \bar{a}$ by a constant $\bar{a}$, which is a stationary point of the potential $V'(\bar{a}) = 0$, centres the excitations $\{\tilde{a}_i\}_{i=1,2,\dots,N}$ around a vacuum. For the model with the Gaussian potential (but an arbitrary measure), the centre of mass decouples. The $SU(N)$ and $U(N)$ loop averages are then simply related as we show in the appendix C. For arbitrary potential the situation is more complicated but still manageable as we demonstrate below.

The ability of operating infinitesimal changes of the potential (28), without breaking the stationarity condition at $\bar{a}$, is the difficult bit to demonstrate. Let us assume the target potential and its deformations to be concave ($V(a) \to +\infty$ for $a \to \pm\infty$) and to have one or many minima $\bar{a}$. The new potential is still polynomial

$$\tilde{V}(\tilde{a}_i) = V(\bar{a} + \tilde{a}_i) = \frac{8\pi^2}{\lambda} \sum_{p=0,2,3,\dots} \tilde{T}_p \tilde{a}_i^p, \tag{56}$$

and parametrised by $\{\bar{a}, \tilde{T}_0, \tilde{T}_2, \tilde{T}_3, \dots\}$. The relations

$$\tilde{T}_1 = \sum_{p=1}^{\infty} p T_p \bar{a}^{p-1} = 0, \qquad \tilde{T}_{p'} = \sum_{p=p'}^{\infty} T_p \binom{p}{p'} \bar{a}^{p-p'}, \qquad p' = 0, 2, 3, \dots, \tag{57}$$

determine $\bar{a} = \bar{a}(T_1, T_2, \dots)$, albeit not uniquely, and $\tilde{T}_{p'} = \tilde{T}_{p'}(T_{p'}, T_{p'+1}, \dots)$ respectively. The effect on Wilson loops is

$$\tilde{\mathbb{W}}(x) = e^{-\bar{a}x} \mathbb{W}(x), \qquad \tilde{\mathbb{W}}(x, y) = e^{-\bar{a}(x+y)} \mathbb{W}(x, y), \tag{58}$$

by the definitions (2) and (3). Matrix averages with tilded symbols are averages of the type (4)

---

[6]The constraints of positivity are very powerful and in conjunction with the loop equations can be used to bootstrap matrix models not solvable otherwise [56].

[7]We are grateful to D. Bykov for this comment.

where $\{\tilde{a}_i\}$ take the place of $\{a_i\}$. The operator (12) undergoes a major change

$$\frac{\delta}{\delta \tilde{V}(x)} \equiv e^{-\bar{a}x}\frac{\delta}{\delta V(x)} = \frac{\lambda}{8\pi^2}\left[\frac{x}{2\tilde{T}_2}\frac{\partial}{\partial \bar{a}} + \sum_{m'=0,2,3,\dots}\left(\frac{(m'+1)x}{2\tilde{T}_2}\tilde{T}_{m'+1} - \frac{x^{m'}}{m'!}\right)\frac{\partial}{\partial \tilde{T}_{m'}}\right]. \tag{59}$$

The proof is based on three lemmas. First, the left equation in (57) gives

$$\frac{\partial \bar{a}}{\partial T_m} = -\frac{8\pi^2}{\lambda}\frac{m\bar{a}^{m-1}}{V''(\bar{a})}. \tag{60}$$

Second, we repeat on the right equation: use (60) with $m \geqslant m' \neq 1$

$$\frac{\partial \tilde{T}_{m'}}{\partial T_m} = \binom{m}{m'}\bar{a}^{m-m'} - \frac{8\pi^2}{\lambda}\frac{m}{V''(\bar{a})}\sum_{p=m'}^{\infty}\binom{p}{m'}(p-m')T_p\,\bar{a}^{p-m'+m-2}, \tag{61}$$

use the inverse of left equation $T_p = \sum_{p'=p}^{\infty}\tilde{T}_{p'}\binom{p'}{p}(-\bar{a})^{p'-p}$

$$\frac{\partial \tilde{T}_{m'}}{\partial T_m} = \binom{m}{m'}\bar{a}^{m-m'} - \frac{8\pi^2}{\lambda}\frac{m\bar{a}^{m-m'-2}}{V''(\bar{a})}\sum_{p=m'}^{\infty}\sum_{p'=p}^{\infty}(-)^{p'-p}\binom{p}{m'}\binom{p'}{p}(p-m')\tilde{T}_{p'}\,\bar{a}^{p'}, \tag{62}$$

swap the sums and resum

$$\frac{\partial \tilde{T}_{m'}}{\partial T_m} = \binom{m}{m'}\bar{a}^{m-m'} - \frac{8\pi^2}{\lambda}\frac{m(m'+1)\bar{a}^{m-1}}{V''(\bar{a})}\tilde{T}_{m'+1}. \tag{63}$$

Third, we begin with (5) and compare to (57)

$$V''(\bar{a}) = \frac{8\pi^2}{\lambda}\sum_{p=2}^{\infty}p(p-1)T_p\bar{a}^{p-2} = \frac{16\pi^2}{\lambda}\tilde{T}_2. \tag{64}$$

Finally, we collect the lemmas and change variables in (12) via the chain rule.

$$\begin{aligned}\frac{\delta}{\delta V(x)} &= -\frac{\lambda}{8\pi^2}\sum_{m=0}^{\infty}\frac{x^m}{m!}\left(\frac{\partial \bar{a}}{\partial T_m}\frac{\partial}{\partial \bar{a}} + \sum_{m'=0,2,3,\dots}\frac{\partial \tilde{T}_{m'}}{\partial T_m}\frac{\partial}{\partial \tilde{T}_{m'}}\right)\\ &= e^{\bar{a}x}\left[\frac{x}{V''(\bar{a})}\frac{\partial}{\partial \bar{a}} + \sum_{m'=0,2,3,\dots}\left(\frac{(m'+1)x}{V''(\bar{a})}\tilde{T}_{m'+1} - \frac{\lambda}{8\pi^2}\frac{x^{m'}}{m'!}\right)\frac{\partial}{\partial \tilde{T}_{m'}}\right]\\ &= e^{\bar{a}x}\frac{\lambda}{8\pi^2}\left[\frac{x}{2\tilde{T}_2}\frac{\partial}{\partial \bar{a}} + \sum_{m'=0,2,3,\dots}\left(\frac{(m'+1)x}{2\tilde{T}_2}\tilde{T}_{m'+1} - \frac{x^{m'}}{m'!}\right)\frac{\partial}{\partial \tilde{T}_{m'}}\right].\end{aligned} \tag{65}$$

This ends the proof of (59).

We are ready to insert (56) and (58) into (27). We pull out $e^{-\bar{a}x}$ on both sides thanks to the definition (59) and the shift property

$$\tilde{V}'\left(\frac{d}{dx} - \bar{a}\right)\left(e^{\bar{a}x}\tilde{\mathbb{W}}(x)\right) = e^{\bar{a}x}\tilde{V}'\left(\frac{d}{dx}\right)\tilde{\mathbb{W}}(x), \tag{66}$$

and derive

$$\tilde{V}'\left(\frac{d}{dx}\right)\tilde{\mathbb{W}}(x) = \frac{1}{2}\int_{-\infty}^{+\infty}\frac{d\omega}{2\pi}\hat{\gamma}(\omega)\int_0^x ds\left[\tilde{\mathbb{W}}(s-i\omega)\tilde{\mathbb{W}}(x-s+i\omega) + \frac{\tilde{\mathbb{W}}(s-i\omega, x-s+i\omega)}{N^2}\right], \tag{67}$$

where the relations (19) and (28) still hold

$$\tilde{V}'\left(\frac{d}{dx}\right) = \frac{8\pi^2}{\lambda}\sum_{n=2}^{\infty} n\tilde{T}_n\left(\frac{d}{dx}\right)^{n-1}, \qquad \tilde{\mathbb{W}}(x,y) = \frac{\delta\tilde{\mathbb{W}}(y)}{\delta\tilde{V}(x)}. \tag{68}$$

The reader should be aware that this *not* an independent loop equation. However there is more than a formal distinction with (27): the averages in (67) are written in matrix variables that ensure $\tilde{V}'(0) \propto \tilde{T}_1 = 0$, but artificially complicate the insertion rule (59).

## 3.2 Power-series Ansatz

The equation is tractable analytically in the limit of large $N$ and small $\lambda$ with the Ansatz

$$\tilde{\mathbb{W}}(x) = \tilde{c}_{000} + \sum_{n=0}^{\infty}\sum_{\ell=2n}^{\infty}\sum_{m=1}^{2\ell} \frac{\tilde{c}_{n,\ell,m}}{(16\pi^2)^{\ell}} N^{-2n}\lambda^{\ell}x^m, \qquad \tilde{c}_{000} = 1. \tag{69}$$

The rationale behind the index ranges becomes clear a posteriori. The power of $\lambda$ puts an upper bound on that of $x$ due to diagrammatical arguments. In (4) written in tilded variables, the limit $\lambda \to 0$ drives eigenvalues $\tilde{a}_i \to 0$ towards the minimum of the potential. One can expand the "interacting" potential $\exp(8\pi^2\lambda^{-1}N\sum_i\sum_{n=3}^{\infty}\tilde{T}_n\tilde{a}_i^n)$ and the rest of the integrand (including a test measure, e.g. $\mu(a) = a^2 + a^4$), then average polynomials of $\tilde{a}_i's$ with the Gaussian weight $\exp(8\pi^2\lambda^{-1}N\tilde{T}_2\sum_i\tilde{a}_i^2)$. This is doable analytically for each term of the series in $\lambda$, which is nothing but some Feynman diagrams (integrals) of a quantum field theory (matrix model) for $N$ scalar fields $\tilde{a}_i$ in zero dimensions. Each interaction vertex (labeled by $n \geq 3$) brings a power of $\lambda^{-1}$ and each propagator (after Gaussian integration) a $\lambda$. When one throws $x$ in by choosing $f(\tilde{a}_1,\ldots,\tilde{a}_N) = \sum_i\exp(x\tilde{a}_i)$, the Gaussian integral of any monomial is proportional to $\lambda^{\ell}x^m$ with $m \leq 2n$. This further restricts to $m = 2\ell$ for Gaussian Hermitian models ($\tilde{T}_{n\geq 3} = 0$ and $\mu(a) = a^2$).

We also put forward

$$\tilde{\mathbb{W}}(x,y) = \sum_{n=0}^{\infty}\sum_{\ell=2n+1}^{\infty}\sum_{m=1}^{2\ell-1}\sum_{m'=1}^{2\ell-m} \frac{\tilde{d}_{n,\ell,m,m'}}{(16\pi^2)^{\ell}} N^{-2n}\lambda^{\ell}x^m y^{m'}. \tag{70}$$

Necessary conditions for the symmetry $x \leftrightarrow y$ are $\tilde{d}_{n,\ell,m,m'} = \tilde{d}_{n,\ell,m',m}$ and $m + m' \leq 2\ell$.

The coefficients $\{\tilde{c}_{n,\ell,m}, \tilde{d}_{n,\ell,m,m'}\}$ in (69) and (70) constitute the maximal set required by arbitrary measure and potential; the simpler these are, the more of the coefficients vanish. It is often useful to pad our formulas with extra coefficients though, namely those labeled by indices out of the ranges in (69) and (70), for example $\tilde{c}_{0,0,1} = 0$.

The coefficients are related via (68):

$$\tilde{d}_{n,\ell,1,m'} = \frac{\tilde{c}_{n,\ell-1,m'-1}}{\tilde{T}_2} + \sum_{m''=2}^{2\ell-m'+1}(m''+1)\frac{\tilde{T}_{m''+1}}{\tilde{T}_2}\frac{\partial\tilde{c}_{n,\ell-1,m'}}{\partial\tilde{T}_{m''}}, \tag{71}$$

for $m' = 1,\ldots,2\ell-1$ and

$$\tilde{d}_{n,\ell,m,m'} = -\frac{2}{m!}\frac{\partial\tilde{c}_{n,\ell-1,m'}}{\partial\tilde{T}_m}, \tag{72}$$

for $m = 2,\ldots,2\ell-1$ and $m' = 1,\ldots,2\ell-m$. As per our convention, we have $\tilde{d}_{0,1,1,1} = \tilde{T}_2^{-1}$ for example. The validity of the Ansatz relies on the stationary point being extremal, i.e. $V''(\bar{a}) = \tilde{V}''(0) \propto \tilde{T}_2 \neq 0$. The apparent asymmetry between (71) and (72) arises from the

partition operated between $\tilde{T}_1 = 0$ and $\{\tilde{T}_n\}_{n=0,2,3,\dots}$ and placing $x$ and $y$ as in (68). The symmetry $\tilde{d}_{n,\ell,m,m'} = \tilde{d}_{n,\ell,m',m}$ is restored *on the solution* of the loop equation.

The coefficients are sensitive to the measure through its Taylor coefficients

$$\gamma^{(0)} = \gamma(0) = 2, \qquad \gamma^{(n)} = \frac{d^n\gamma}{dx^n}(0) = \int_{-\infty}^{+\infty} \frac{d\omega}{2\pi}(-i\omega)^n\hat{\gamma}(\omega), \qquad n = 2, 4, 6, \dots, \tag{73}$$

the potential through the extremal point $\bar{a}$ and the Taylor coefficients near this point. The indices restrict the dependence:

$$\frac{\partial \tilde{c}_{n,\ell,m}}{\partial \tilde{T}_{m'}} = 0, \qquad m' \geqslant 2\ell - m + 3, \tag{74}$$

therefore the solution (69) and (70) up to the order $N^{-2n_{\max}}\lambda^{\ell_{\max}}$ with $2n_{\max} \leqslant \ell_{\max}$ depends on the finitely-many coefficients $\{\tilde{T}_2, \tilde{T}_3, \dots, \tilde{T}_{2\ell_{\max}+1}\}$ of (56). The bound is directly visible at the level of (67) once one plugs (56) and (69).

## 3.3   Solving algorithm

We build upon the ansätze (69) and (70) to turn the loop equation (67) into a system of algebraic equations for the infinitely-many coefficients $\{\tilde{c}_{n,\ell,m}\}$. We take note of (71) and (72), but use the dependent set $\{\tilde{d}_{n,\ell,m,m'}\}$ too in order not to clutter formulas.

**Planar limit.**   The limit of infinite $N$ washes away the two-point Wilson loop in (67) and most coefficients, save for the subset $\{\tilde{c}_{0,\ell,m}\}$ with $\ell = 1, 2, \dots$ and $m = 1, 2, \dots, 2\ell$. Appendix D details how to spell out (69) in the equation. Equating powers of $\lambda$ and $x$ leads to two systems of equations:

$$\sum_{r=2}^{2\ell+1} r!\,\tilde{T}_r c_{0,\ell,r-1} = 0, \tag{75}$$

spanned by $\ell = 1, 2, \dots$ and

$$\frac{1}{p!}\sum_{r=2}^{2\ell-p+1} r(p+r-1)!\tilde{T}_r\tilde{c}_{0,\ell,p+r-1} \tag{76}$$

$$= \sum_{q=1-p,3-p,\dots,p-1}\,\sum_{m=\frac{p-q-1}{2}}^{2\ell-2}\,\sum_{m'=\frac{p+q-1}{2}}^{2\ell-2}\,\sum_{\ell'=\lceil\frac{m}{2}\rceil}^{\ell-\lceil\frac{m'}{2}\rceil-1}\,\sum_{d=-\frac{p+q-1}{2},-\frac{p+q-1}{2}+2,\dots,\frac{p+q-1}{2}}\binom{m}{\frac{p-q-1}{2}}$$

$$\times \binom{m'}{\frac{p+q-1}{4}+\frac{d}{2}}\binom{m'-\frac{p+q-1}{4}-\frac{d}{2}}{\frac{p+q-1}{4}-\frac{d}{2}}\frac{(-)^{m'-\frac{p+q-1}{4}-\frac{d}{2}}}{\frac{3p-q+1}{4}-\frac{d}{2}}\gamma^{(m+m'-p+1)}\tilde{c}_{0,\ell',m}\tilde{c}_{0,\ell-\ell'-1,m'},$$

by $\ell = 1, 2, \dots$ and $p = 1, \dots, 2\ell - 1$. The sums with $\gamma^{(m+m'-p+1)}$ are constrained to have $m + m' - p$ odd by (73). The terms $\tilde{c}^2$ inherit the non-linearity of the loop equation. There is a solution strategy that handles only linear equations.

Suppose the iteration $\ell^*$ knows the subset $\{\tilde{c}_{0,\ell,m}\}$ with $\ell = 1, 2, \dots, \ell^* - 1$ and $m = 1, 2, \dots, 2\ell$. We recall $\tilde{T}_2 \neq 0$ below (72) and express $\tilde{c}_{0,\ell^*,1}$ in terms of $\{\tilde{c}_{0,\ell^*,m}\}_{m\geqslant 2}$ through (75)

$$\tilde{c}_{0,\ell^*,1} = -\frac{1}{2\tilde{T}_2}\sum_{r=3}^{2\ell^*+1} r!\,\tilde{T}_r\tilde{c}_{0,\ell^*,r-1}. \tag{77}$$

We replace this in (76) and verify that the equations are linear in the coefficients $\{\tilde{c}_{0,\ell^*,m}\}_{m\geqslant 2}$; in particular, these coefficients never appear quadratically in the right-hand side, but linearly and each multiplied by one found at a previous iteration. Moreover, the equations are inhomogeneous due to the terms $\gamma^{(m+m'-p+1)}$, hence the $\{\tilde{c}_{0,\ell^*,m}\}_{m\geqslant 2}$ turn out to be polynomials in the Taylor coefficients of $\gamma(a)$. Standard techniques can efficiently solve (76)[8] and use (77) to calculate $\tilde{c}_{0,\ell^*,1}$.

**Sub-planar corrections.** We apply the same logic to the full equation (67) to find $\{c_{n,\ell,m}\}$ and $\{d_{n,\ell,m,m'}\}$. This spells out a (bulky) system of equations, after keeping powers of $N$ in appendix D and considering the two-point Wilson loop. We prefer however to put together the numerous observations so far and present the algorithmic workflow.

1. The goal is to find the solution of (67) up to order $N^{-2n_{\max}}\lambda^{\ell_{\max}}$ in the form

$$\tilde{\mathbb{W}}(x) = \sum_{n=0}^{n_{\max}} \sum_{\ell=2n}^{\ell_{\max}} \tilde{\mathbb{W}}_{n,\ell}(x) N^{-2n}\lambda^\ell, \qquad \tilde{\mathbb{W}}_{n,\ell}(x) = \delta_{n,0}\delta_{\ell,0} + \sum_{m=1}^{2\ell} \frac{\tilde{c}_{n,\ell,m}}{(16\pi^2)^\ell} x^m, \quad (78)$$

and as a by-product

$$\tilde{\mathbb{W}}(x,y) = \sum_{n=0}^{n_{\max}} \sum_{\ell=2n+1}^{\ell_{\max}} \tilde{\mathbb{W}}_{n,\ell}(x,y) N^{-2n}\lambda^\ell,$$

$$\tilde{\mathbb{W}}_{n,\ell}(x,y) = \sum_{m=1}^{2\ell-1} \sum_{m'=1}^{2\ell-m} \frac{\tilde{d}_{n,\ell,m,m'}}{(16\pi^2)^\ell} x^m y^{m'}. \quad (79)$$

We remind the measure factor obeys $\gamma(0) = 2$ and approximate the potential to a finite-degree polynomial

$$\tilde{V}(x) = \frac{8\pi^2}{\lambda} \sum_{n=0}^{2\ell_{\max}+1} \tilde{T}_n x^n, \qquad \tilde{T}_1 = 0, \qquad \tilde{T}_2 \neq 0, \quad (80)$$

thanks to the observation below (74). The coefficients can take numerical values from the get-go when the target is the planar limit ($n_{\max} = 0$) of (78). They cannot otherwise, for the sake of the step 2 which extracts (79) from (78) by differentiation.

2. We compute the coefficients in (78) and (79) iteratively. We loop over $n^* = 0, 1, \ldots, n_{\max}$ and $\ell^* = 2n^*, 2n^*+1, \ldots, \ell_{\max}$[9] to find the subsets $\{\tilde{c}_{n^*,\ell^*,m}\}_m$ and $\{\tilde{d}_{n^*,\ell^*,m,m'}\}_{m,m'}$. The iteration starts by collecting terms $N^{-2n^*}\lambda^{\ell^*-1}$ in (67)

$$\tilde{V}'\left(\frac{d}{dx}\right)\tilde{\mathbb{W}}_{n^*,\ell^*}(x) = \frac{1}{2}\int_{-\infty}^{+\infty} \frac{d\omega}{2\pi}\hat{\gamma}(\omega)\int_0^x ds\left[\sum_{p=0}^{n^*}\sum_{q=0}^{\ell^*-1}\tilde{\mathbb{W}}_{p,q}(s-i\omega)\right.$$

$$\left.\times\tilde{\mathbb{W}}_{n^*-p,\ell^*-q-1}(x-s+i\omega) + \frac{1}{N^2}\tilde{\mathbb{W}}_{n^*-1,\ell^*-1}(s-i\omega,x-s+i\omega)\right]. \quad (81)$$

The last term exists at sub-planar level ($n^* \geqslant 1$) and involves the coefficients $\{\tilde{d}_{n^*-1,\ell^*-1,m,m'}\}_{m,m'}$ found in a previous iteration. The integrand is a polynomial in $\omega$,

---

[8] Existence and uniqueness of the solution fall back on that of the loop equation.
[9] The case $n^* = \ell^* = 0$ is trivial.

$s$ and $x$; each monomial can be integrated via Fourier transform (73). The integro-differential equation (81) turns into a polynomial equation which must hold regardless of the value of $x$. The coefficients of $x$-powers are proportional to one of $\{\tilde{c}_{n^*,\ell^*,m}\}_m$. Therefore equating homonymous powers of $x$ writes a finite system of linear equations for $\{\tilde{c}_{n^*,\ell^*,m}\}_m$. Once the solution is found, the iteration ends with the computation of $\{\tilde{d}_{n^*,\ell^*,m,m'}\}_{m,m'}$ using (71) and (72).

3. At this stage one knows (78) and (79) for an arbitrary potential (80) with extremal point in the origin. The analysis of section 3.1 comes in handy when we want to translate it to a generic point $\bar{a}$. We translate (80) according to (56)

$$V(a) = \tilde{V}(a - \bar{a}) = \frac{8\pi^2}{\lambda} \sum_{p=0}^{2\ell_{\max}+1} T_p a^p \,, \tag{82}$$

$$T_p = \sum_{p'=p}^{2\ell_{\max}+1} \tilde{T}_{p'} \binom{p'}{p} (-\bar{a})^{p'-p} \,, \qquad p = 0, 1, \dots, 2\ell_{\max} + 1 \,. \tag{83}$$

The solution of (27) for this potential is given by (58)

$$\mathbb{W}(x) = e^{\bar{a}x} \tilde{\mathbb{W}}(x), \qquad \mathbb{W}(x,y) = e^{\bar{a}(x+y)} \tilde{\mathbb{W}}(x,y) \,. \tag{84}$$

4. We assign numeric values to $\{\tilde{T}_n\}_{n=1,\dots,\ell_{\max}}$ in order to specialise (82) to a target potential. The task is already part of step 1 in the special case mentioned therein. Notice different potentials correspond to the same solution as long as they share the same shape sufficiently near the extremal point.

# 4 Weak coupling: Applications

We run the algorithm in section 3.3 for a selection of matrix models, perform checks with the literature and derive new results for Wilson loops in SYM theories. We remind that (78) solves (67) with potential $\tilde{V}'(0) = 0$. Step 3 of the algorithm converts the solution to that (84) associated to the shifted potential $V(a) = \tilde{V}(a - \bar{a})$. The distinction between tilded and untilded variables is immaterial when $\bar{a} = 0$ and the focus is on the planar limit.

## 4.1 Models with Vandermonde measure

The Vandermonde measure $\mu(a) = a^2$ maps to a constant $\gamma(a) = 2$. This brings massive simplifications: the Dirac delta $\hat{\gamma}(\omega) = 4\pi\delta(\omega)$ trivialises one integral in the loop equation and all derivatives $\gamma^{(n)} = 2\delta_{n,0}$ in (73) drop out of step 1. The planar coefficients of (78) up to $\lambda^3$ are

$$\tilde{c}_{0,1,1} = -\frac{3\tilde{T}_3}{2\tilde{T}_2^2}, \qquad \tilde{c}_{0,1,2} = \frac{1}{2\tilde{T}_2}, \qquad \tilde{c}_{0,2,1} = \frac{-54\tilde{T}_3^3 + 72\tilde{T}_2\tilde{T}_3\tilde{T}_4 - 20\tilde{T}_2^2\tilde{T}_5}{4\tilde{T}_2^5},$$

$$\tilde{c}_{0,2,2} = \frac{18\tilde{T}_3^2 - 8\tilde{T}_2\tilde{T}_4}{4\tilde{T}_2^4}, \qquad \tilde{c}_{0,2,3} = -\frac{\tilde{T}_3}{\tilde{T}_2^3}, \qquad \tilde{c}_{0,2,4} = \frac{1}{12\tilde{T}_2^2},$$

$$\tilde{c}_{0,3,1} = \frac{-3888\tilde{T}_3^5 + 9936\tilde{T}_2\tilde{T}_3^3\tilde{T}_4 - 4320\tilde{T}_2^2\tilde{T}_3(\tilde{T}_4^2 + \tilde{T}_3\tilde{T}_5)}{16\tilde{T}_2^8}$$

$$+ \frac{1440\tilde{T}_2^3(\tilde{T}_4\tilde{T}_5 + \tilde{T}_3\tilde{T}_6) - 280\tilde{T}_2^4\tilde{T}_7}{16\tilde{T}_2^8} \,,$$

$$\tilde{c}_{0,3,2} = \frac{3888\tilde{T}_3^4 - 6480\tilde{T}_2\tilde{T}_3^2\tilde{T}_4 + 2160\tilde{T}_2^2\tilde{T}_3\tilde{T}_5 + 864\tilde{T}_2^2\tilde{T}_4^2 - 360\tilde{T}_2^3\tilde{T}_6}{48\tilde{T}_2^7}, \tag{85}$$

$$\tilde{c}_{0,3,3} = \frac{144\tilde{T}_3(-\tilde{T}_3^2 + \tilde{T}_2\tilde{T}_4) - 30\tilde{T}_2^2\tilde{T}_5}{8\tilde{T}_2^6},$$

$$\tilde{c}_{0,3,4} = \frac{27\tilde{T}_3^2 - 9\tilde{T}_2\tilde{T}_4}{12\tilde{T}_2^5}, \qquad \tilde{c}_{0,3,5} = -\frac{3\tilde{T}_3}{16\tilde{T}_2^4}, \qquad \tilde{c}_{0,3,6} = \frac{1}{144\tilde{T}_2^3},$$

and those at genus one

$$\tilde{c}_{1,2,1} = \frac{-27\tilde{T}_3^3 + 48\tilde{T}_2\tilde{T}_3\tilde{T}_4 - 20\tilde{T}_2^2\tilde{T}_5}{8\tilde{T}_2^5}, \qquad \tilde{c}_{1,2,2} = \frac{9\tilde{T}_3^2 - 8\tilde{T}_2\tilde{T}_4}{8\tilde{T}_2^4},$$

$$\tilde{c}_{1,2,3} = -\frac{\tilde{T}_3}{4\tilde{T}_2^3}, \qquad \tilde{c}_{1,2,4} = \frac{1}{24\tilde{T}_2^2},$$

$$\tilde{c}_{1,3,1} = \frac{1}{16\tilde{T}_2^8}\Big[ -3402\tilde{T}_3^5 + 10044\tilde{T}_2\tilde{T}_3^3\tilde{T}_4 - 5220\tilde{T}_2^2\tilde{T}_3^2\tilde{T}_5$$

$$+ 432\tilde{T}_2^2\tilde{T}_3(-12\tilde{T}_4^2 + 5\tilde{T}_2\tilde{T}_6) + 80\tilde{T}_2^3(27\tilde{T}_4\tilde{T}_5 - 7\tilde{T}_2\tilde{T}_7)\Big], \tag{86}$$

$$\tilde{c}_{1,3,2} = \frac{3402\tilde{T}_3^4 - 7020\tilde{T}_2\tilde{T}_3^2\tilde{T}_4 + 2880\tilde{T}_2^2\tilde{T}_3\tilde{T}_5 - 720\tilde{T}_2^2(-2\tilde{T}_4^2 + \tilde{T}_2\tilde{T}_6)}{48\tilde{T}_2^7},$$

$$\tilde{c}_{1,3,3} = \frac{-126\tilde{T}_3^3 + 156\tilde{T}_2\tilde{T}_3\tilde{T}_4 - 40\tilde{T}_2^2\tilde{T}_5}{8\tilde{T}_2^6}, \qquad \tilde{c}_{1,3,4} = \frac{177\tilde{T}_3^2 - 60\tilde{T}_2\tilde{T}_4}{48\tilde{T}_2^5},$$

$$\tilde{c}_{1,3,5} = -\frac{\tilde{T}_3}{4\tilde{T}_2^4}, \qquad \tilde{c}_{1,3,6} = \frac{1}{72\tilde{T}_2^3}.$$

The Vandermonde measure is the hallmark of the Hermitian model and plenty of closed-form formulas populate the literature. The two-point resolvent was considered in topological expansion [57] and finite $N$ [58]. We test the solution with the small-$\lambda$ expansion of (87) and (88) below.

The genus-zero and -one resolvents are solvable in closed form [6]. The inverse Laplace transform is worked out in [57], for example: for sixth-order even potentials

$$\mathbb{W}(x) = \frac{2}{\mu x}I_1(\mu x) + \frac{3\pi^2}{\lambda}\left(4\tilde{T}_4 + \frac{15}{2}\tilde{T}_6\mu^2\right)\frac{\mu^3}{x}I_3(\mu x) + \frac{15\pi^2}{2\lambda}\tilde{T}_6\frac{\mu^5}{x}I_5(\mu x) \tag{87}$$

$$+ \frac{\lambda}{32\pi^2N^2}\left[\frac{x^2I_2(\mu x)}{3(2\tilde{T}_2 + 6\tilde{T}_4\mu^2 + \frac{45}{4}\tilde{T}_6\mu^4)} - \frac{(4\tilde{T}_4\mu + 15\tilde{T}_6\mu^3)xI_1(\mu x)}{(2\tilde{T}_2 + 6\tilde{T}_4\mu^2 + \frac{45}{4}\tilde{T}_6\mu^4)^2}\right] + O(N^{-4}),$$

and for general even potentials

$$\mathbb{W}(x,y) = \frac{\mu xy}{2(x+y)}\left[I_0(\mu x)I_1(\mu y) + I_0(\mu y)I_1(\mu x)\right] + O(N^{-2}), \tag{88}$$

where $I_n$ is the modified Bessel function of the first kind. The coupling dependence enters the branch-cut endpoints $[-\mu, \mu]$, with $\mu$ the solution of the algebraic equation

$$\text{Res}\left(\frac{V'(\frac{1}{z})}{z^2\sqrt{1-\mu^2z^2}}, z=0\right) = \frac{16\pi^2}{\lambda}\sum_{p=2,4,\dots}\frac{\Gamma\left(\frac{p+1}{2}\right)}{\sqrt{\pi}\,\Gamma\left(\frac{p}{2}\right)}\tilde{T}_p\mu^p = 2. \tag{89}$$

The correct solution approximates the branch point of the Wigner semicircle distribution

$$\mu = \frac{1}{2\pi}\sqrt{\frac{\lambda}{\tilde{T}_2}}\left[1 + \sum_{\ell=1}^{\infty}\mu_\ell\left(\frac{\lambda}{16\pi^2\tilde{T}_2^2}\right)^\ell\right], \tag{90}$$

with the corrections

$$\mu_1 = -3\tilde{T}_4, \tag{91}$$

$$\mu_2 = \frac{1}{2}(63\tilde{T}_4^2 - 30\tilde{T}_2\tilde{T}_6), \tag{92}$$

$$\mu_3 = \frac{1}{2}(-891\tilde{T}_4^3 + 810\tilde{T}_2\tilde{T}_4\tilde{T}_6 - 140\tilde{T}_2^2\tilde{T}_8), \tag{93}$$

$$\mu_4 = \frac{15}{8}(3861\tilde{T}_4^4 - 5148\tilde{T}_2\tilde{T}_4^2\tilde{T}_6 + 660\tilde{T}_2^2\tilde{T}_6^2 + 1232\tilde{T}_2^2\tilde{T}_4\tilde{T}_8 - 168\tilde{T}_2^3\tilde{T}_{10}). \tag{94}$$

## 4.2 Models with Gaussian potential

The next-to-simplest models feature the weight

$$V(a) = \frac{8\pi^2}{\lambda}a^2, \tag{95}$$

alongside the smooth measure (73). We list the non-zero planar coefficients up to $\lambda^6$

$$\tilde{c}_{0,1,2} = \frac{1}{2}, \qquad \tilde{c}_{0,2,2} = \frac{\gamma^{(2)}}{4}, \qquad \tilde{c}_{0,2,4} = \frac{1}{12}, \qquad \tilde{c}_{0,3,2} = \frac{6(\gamma^{(2)})^2 + 5\gamma^{(4)}}{48},$$

$$\tilde{c}_{0,3,4} = \frac{\gamma^{(2)}}{12}, \qquad \tilde{c}_{0,3,6} = \frac{1}{144}, \qquad \tilde{c}_{0,4,2} = \frac{18(\gamma^{(2)})^3 + 45\gamma^{(2)}\gamma^{(4)} + 7\gamma^{(6)}}{288},$$

$$\tilde{c}_{0,4,4} = \frac{12(\gamma^{(2)})^2 + 7\gamma^{(4)}}{192}, \qquad \tilde{c}_{0,4,6} = \frac{\gamma^{(2)}}{96}, \qquad \tilde{c}_{0,4,8} = \frac{1}{2880},$$

$$\tilde{c}_{0,5,2} = \frac{180(\gamma^{(2)})^4 + 900(\gamma^{(2)})^2\gamma^{(4)} + 255(\gamma^{(4)})^2 + 280\gamma^{(2)}\gamma^{(6)} + 21\gamma^{(8)}}{5760},$$

$$\tilde{c}_{0,5,4} = \frac{360(\gamma^{(2)})^3 + 630\gamma^{(2)}\gamma^{(4)} + 77\gamma^{(6)}}{8640}, \qquad \tilde{c}_{0,5,6} = \frac{20(\gamma^{(2)})^2 + 9\gamma^{(4)}}{1920}, \tag{96}$$

$$\tilde{c}_{0,5,8} = \frac{\gamma^{(2)}}{1440}, \qquad \tilde{c}_{0,5,10} = \frac{1}{86400},$$

$$\tilde{c}_{0,6,2} = \frac{1}{172800}\Big[2700(\gamma^{(2)})^5 + 22500(\gamma^{(2)})^3\gamma^{(4)}$$
$$+ 10500(\gamma^{(2)})^2\gamma^{(6)} + 4550\gamma^{(4)}\gamma^{(6)} + 225\gamma^{(2)}(85(\gamma^{(4)})^2$$
$$+ 7\gamma^{(8)}) + 66\gamma^{(10)}\Big],$$

$$\tilde{c}_{0,6,4} = \frac{450(\gamma^{(2)})^4 + 1575(\gamma^{(2)})^2\gamma^{(4)} + 345(\gamma^{(4)})^2 + 385\gamma^{(2)}\gamma^{(6)} + 24\gamma^{(8)}}{17280},$$

$$\tilde{c}_{0,6,6} = \frac{900(\gamma^{(2)})^3 + 1215\gamma^{(2)}\gamma^{(4)} + 122\gamma^{(6)}}{103680}, \qquad \tilde{c}_{0,6,8} = \frac{30(\gamma^{(2)})^2 + 11\gamma^{(4)}}{34560},$$

$$\tilde{c}_{0,6,10} = \frac{\gamma^{(2)}}{34560}, \qquad \tilde{c}_{0,6,12} = \frac{1}{3628800},$$

the $N^{-2}$-coefficients

$$\tilde{c}_{1,2,2} = -\frac{\gamma^{(2)}}{4}, \qquad \tilde{c}_{1,2,4} = \frac{1}{24}, \qquad \tilde{c}_{1,3,2} = -\tilde{c}_{0,3,2},$$

$$\tilde{c}_{1,3,4} = -\frac{\gamma^{(2)}}{12}, \qquad \tilde{c}_{1,3,6} = \frac{1}{72},$$

$$\tilde{c}_{1,4,2} = -\tilde{c}_{0,4,2}, \qquad \tilde{c}_{1,4,4} = \frac{-18(\gamma^{(2)})^2 - 5\gamma^{(4)}}{192}, \qquad \tilde{c}_{1,4,8} = \frac{1}{576},$$

$$\tilde{c}_{1,5,2} = \frac{-36(\gamma^{(2)})^4 - 180(\gamma^{(2)})^2\gamma^{(4)} - 15(\gamma^{(4)})^2 - 56\gamma^{(2)}\gamma^{(6)} - 3\gamma^{(8)}}{1152},$$

$$\tilde{c}_{1,5,4} = \frac{-126(\gamma^{(2)})^3 - 135\gamma^{(2)}\gamma^{(4)} - 7\gamma^{(6)}}{1728}, \qquad \tilde{c}_{1,5,6} = \frac{-3(\gamma^{(2)})^2 + \gamma^{(4)}}{288}, \qquad (97)$$

$$\tilde{c}_{1,5,8} = \frac{\gamma^{(2)}}{576}, \qquad \tilde{c}_{1,5,10} = \frac{1}{8640},$$

$$\tilde{c}_{1,6,2} = \frac{1}{6912}\Big[-108(\gamma^{(2)})^5 - 900(\gamma^{(2)})^3\gamma^{(4)} - 420(\gamma^{(2)})^2\gamma^{(6)}$$
$$+70\gamma^{(4)}\gamma^{(6)} - 45\gamma^{(2)}(5(\gamma^{(4)})^2 + \gamma^{(8)})\Big],$$

$$\tilde{c}_{1,6,4} = \frac{-684(\gamma^{(2)})^4 - 1620(\gamma^{(2)})^2\gamma^{(4)} + 15(\gamma^{(4)})^2 - 224\gamma^{(2)}\gamma^{(6)} + 3\gamma^{(8)}}{13824},$$

$$\tilde{c}_{1,6,6} = \frac{-288(\gamma^{(2)})^3 - 9\gamma^{(2)}\gamma^{(4)} + 35\gamma^{(6)}}{20736}, \qquad \tilde{c}_{1,6,8} = \frac{12(\gamma^{(2)})^2 + 17\gamma^{(4)}}{13824},$$

$$\tilde{c}_{1,6,10} = \frac{7\gamma^{(2)}}{34560}, \qquad \tilde{c}_{1,6,12} = \frac{1}{207360},$$

the $N^{-4}$-coefficients

$$\tilde{c}_{2,4,4} = \frac{3(\gamma^{(2)})^2 - \gamma^{(4)}}{96}, \qquad \tilde{c}_{2,4,6} = -\frac{\gamma^{(2)}}{96}, \qquad \tilde{c}_{2,4,8} = \frac{1}{1920},$$

$$\tilde{c}_{2,5,2} = \frac{-30(\gamma^{(4)})^2 - \gamma^{(8)}}{960}, \qquad \tilde{c}_{2,5,4} = \frac{90(\gamma^{(2)})^3 + 15\gamma^{(2)}\gamma^{(4)} - 14\gamma^{(6)}}{2880},$$

$$\tilde{c}_{2,5,6} = -\frac{47\gamma^{(4)}}{5760}, \qquad \tilde{c}_{2,5,8} = -\frac{7\gamma^{(2)}}{2880}, \qquad \tilde{c}_{2,5,10} = \frac{23}{172800},$$

$$\tilde{c}_{2,6,2} = \frac{-2250\gamma^{(2)}(\gamma^{(4)})^2 - 1050\gamma^{(4)}\gamma^{(6)} - 75\gamma^{(2)}\gamma^{(8)} - 11\gamma^{(10)}}{28800}, \qquad (98)$$

$$\tilde{c}_{2,6,4} = \frac{540(\gamma^{(4)})^2 + 600(\gamma^{(2)})^2\gamma^{(4)} - 485(\gamma^{(4)})^2 - 140\gamma^{(2)}\gamma^{(6)} - 37\gamma^{(8)}}{23040},$$

$$\tilde{c}_{2,6,6} = \frac{270(\gamma^{(2)})^3 - 480\gamma^{(2)}\gamma^{(4)} - 91\gamma^{(6)}}{34560}, \qquad \tilde{c}_{2,6,8} = \frac{-210(\gamma^{(2)})^2 - 77\gamma^{(4)}}{69120},$$

$$\tilde{c}_{2,6,10} = \frac{-7\gamma^{(2)}}{69120}, \qquad \tilde{c}_{2,6,12} = \frac{7}{518400},$$

and the $N^{-6}$-coefficients

$$\tilde{c}_{3,6,6} = \frac{45\gamma^{(2)}(-(\gamma^{(2)})^2 + \gamma^{(4)}) - 4\gamma^{(6)}}{17280}, \qquad \tilde{c}_{3,6,8} = \frac{3(\gamma^{(2)})^2 - \gamma^{(4)}}{2304},$$

$$\tilde{c}_{3,6,10} = -\frac{\gamma^{(2)}}{7680}, \qquad \tilde{c}_{3,6,12} = \frac{1}{322560}. \qquad (99)$$

## 4.3 Circular Wilson loop

Supersymmetric localisation [29] calculates the expectation value of the circular Wilson loop on $S^4$ in $\mathcal{N} \geqslant 2$ SYM theories as the matrix-model operator $\mathbb{W}(2\pi)$ in (2). Here one identifies $N$ and $\lambda$ with that of the gauge group $U(N)$ and with the 't Hooft coupling respectively. The potential

$$V(a) = \frac{8\pi^2}{\lambda}a^2, \qquad (100)$$

is generated by the conformal mass of the vector-multiplet scalar in the definition of the Wilson loop, while the measure comprises the exact one-loop determinants of the field fluctuations near the localisation locus.

### 4.3.1  $\mathcal{N} = 4$ SYM

The maximally-symmetric case is captured by the simplest zero-dimensional dynamics with

$$\mu(a) = a^2 . \tag{101}$$

This falls within the applicability range of sections 4.1 and 4.2. Here we set $\tilde{T}_2 = 1$, $\tilde{T}_{n \geqslant 3} = 0$ due to (100) and $\gamma^{(n)} = 2\delta_{n,0}$ due to (101). We reproduce the weak-coupling expansion of (106) and also the $N^{-2}$-term of (107)[10] below.

The planar loop equation is solvable at finite coupling. The Wilson loop appears under a derivative and a convolution[11]

$$\mathbb{W}'(x) = \frac{\lambda}{16\pi^2} \int_0^x du \, \mathbb{W}(u)\mathbb{W}(x-u), \tag{102}$$

therefore it converts to algebraic form by Laplace transform. Since the resolvent (7) coincides with $\omega(s) = \mathcal{L}_s(\mathbb{W}(x))$, (102) maps to nothing but the equation for the resolvent

$$s\omega(s) - 1 = \frac{\lambda}{16\pi^2}\omega^2(s). \tag{103}$$

The correct solution with the decay $\omega(s) \sim 1/s$ at infinity is

$$\omega(s) = \frac{8\pi^2}{\lambda}\left( s - \sqrt{s^2 - \frac{\lambda}{4\pi^2}} \right), \tag{104}$$

and leads to the famous Wigner semicircle distribution. We transform[12]

$$\mathbb{W}(x) = \mathcal{L}_x^{-1}(\omega(s)) = \frac{4\pi}{x\sqrt{\lambda}}I_1\left( \frac{x\sqrt{\lambda}}{2\pi} \right), \tag{105}$$

and recover the planar circular Wilson loop [27]

$$\mathbb{W}(2\pi) = \frac{2}{\sqrt{\lambda}}I_1\left( \sqrt{\lambda} \right). \tag{106}$$

We recall that this gets replaced by [28][13]

$$\mathbb{W}(x) = \frac{1}{N}e^{\frac{\lambda x^2}{32\pi^2 N}}L_{N-1}^{(1)}\left( -\frac{\lambda x^2}{16\pi^2 N} \right), \tag{107}$$

at finite $N$, where $L_n^{(\alpha)}$ is a generalised Laguerre polynomial.

In general, loop equations compute the genus-zero $n$-point resolvents [12, 63, 64], hence the Wilson loops upon inverse Laplace transforms [57, 65]. High-genus expansion are also available [66] in scaling limits [58]. We reproduce the series of

$$\mathbb{W}(x,y) = \frac{xy\sqrt{\lambda}}{4\pi(x+y)}I_0\left( \frac{x\sqrt{\lambda}}{2\pi} \right)I_1\left( \frac{y\sqrt{\lambda}}{2\pi} \right) + \frac{xy^2\lambda^{3/2}}{48(2\pi)^3 N^2}I_1\left( \frac{x\sqrt{\lambda}}{2\pi} \right)I_2\left( \frac{y\sqrt{\lambda}}{2\pi} \right)$$
$$+ \frac{xy^2\lambda^2}{192(2\pi)^4 N^2}\sum_{k=0,1}\left[ yI_k\left( \frac{x\sqrt{\lambda}}{2\pi} \right)I_{k+2}\left( \frac{y\sqrt{\lambda}}{2\pi} \right) + \frac{x}{2}I_{k+1}\left( \frac{x\sqrt{\lambda}}{2\pi} \right)I_{k+1}\left( \frac{y\sqrt{\lambda}}{2\pi} \right) \right]$$
$$+ (x \leftrightarrow y) + O(N^{-4}). \tag{108}$$

---

[10]An expansion algorithm is presented in [59] and appendix B of [60].

[11]$\mathbb{W}(x)$ maps to the $k$-wound circular Wilson loop, with $k \in \mathbb{N}$ analytically continued to $k = x/(2\pi) \in \mathbb{R}$. Upon a rescaling $x$ and $u$, the relevant parameter in (102) is actually $\lambda x^2$. This matches the gauge-theory dependency of the $k$-wound loop, an example of the coupling-rescaling property of loops on $S^2$ [61, 62].

[12]The series of (104) for $s \to \infty$ matches that of (105) for $\lambda \to 0$.

[13]Multiplying this by $\exp(-\frac{\lambda x^2}{32\pi^2 N^2})$ is the result valid for the $SU(N)$ gauge group.

The rest of the section is devoted to derive the planar order (118) below in a way neater, although less general, than [58]. We begin with (3)

$$\mathbb{W}(x, y) = \left\langle \sum_{i,j} e^{xa_i} e^{ya_j} \right\rangle - N^2 \mathbb{W}(x)\mathbb{W}(y). \tag{109}$$

The second term is evaluated with (107) whereas the first one calls for some ingenuity. Taking inspiration from the coincident Wilson loops [28, 58, 60], we represent it as (4)

$$\left\langle \sum_{i,j} e^{xa_i} e^{ya_j} \right\rangle = \frac{1}{Z} \int_{-\infty}^{+\infty} \prod_i da_i \prod_{i<j} (a_i - a_j)^2 \sum_{i,j} e^{xa_i} e^{ya_j} e^{-\frac{8\pi^2 N}{\lambda} \sum_i a_i^2}, \tag{110}$$

separate $N$ "diagonal" and $N(N-1)$ "non-diagonal" terms

$$\frac{1}{Z} \int_{-\infty}^{+\infty} \prod_i da_i \prod_{i<j} (a_i - a_j)^2 \left[ N e^{a_1(x+y)} + N(N-1) e^{a_1 x + a_2 y} \right] e^{-\frac{8\pi^2 N}{\lambda} \sum_i a_i^2}, \tag{111}$$

and rescale $a_i \to \lambda a_i / (8\pi^2 N)$

$$N\mathbb{W}(x + y) + \frac{N(N-1)}{Z'} \int_{-\infty}^{+\infty} \prod_i da_i \prod_{i<j} (a_i - a_j)^2 e^{\sqrt{\frac{\lambda}{8\pi^2 N}}(a_1 x + a_2 y)} e^{-\sum_i a_i^2}. \tag{112}$$

We apply the method of orthogonal polynomials [28]. The Hermite polynomials

$$P_n(a) = \frac{H_n(a)}{\sqrt{2^n n! \sqrt{\pi}}}, \tag{113}$$

are defined by the recurrence relation $H_0(a) = 1$ and $H_{n+1}(a) = 2xH_n(a) - H'_n(a)$. We integrate over the $N - 2$ eigenvalues that do not appear explicitly in (112)

$$N\mathbb{W}(x + y) + \int_{-\infty}^{+\infty} da\,da' \sum_{i,j=0}^{N-1} \left[ P_i^2(a) P_j^2(a') - P_i(a) P_j(a) P_i(a') P_j(a') \right] e^{\sqrt{\frac{\lambda}{8\pi^2 N}}(ax + a'y)} e^{-a^2 - a'^2}. \tag{114}$$

The orthogonality relation generalises to the formula in Appendix A of [58]

$$\int_{-\infty}^{+\infty} da\, P_i(a) P_j(a) e^{-\left(a - \frac{x}{4\pi}\sqrt{\frac{\lambda}{2N}}\right)^2} = \sqrt{\frac{j!}{i!} \left(\frac{\lambda x^2}{16\pi^2 N}\right)^{i-j}} L_j^{(i-j)}\left(-\frac{\lambda x^2}{16\pi^2 N}\right). \tag{115}$$

Using this and recalling (107), (114) becomes

$$e^{\frac{\lambda(x+y)^2}{32\pi^2 N}} L_{N-1}^{(1)}\left(-\frac{\lambda(x+y)^2}{16\pi^2 N}\right) + e^{\frac{\lambda(x^2+y^2)}{32\pi^2 N}} \sum_{i=1}^{N-1} \sum_{j=0}^{i-1} \left[ L_i\left(-\frac{\lambda x^2}{16\pi^2 N}\right) L_j\left(-\frac{\lambda y^2}{16\pi^2 N}\right) \right.$$
$$\left. - \frac{j!}{i!} \left(\frac{\lambda xy}{16\pi^2 N}\right)^{i-j} L_j^{(i-j)}\left(-\frac{\lambda x^2}{16\pi^2 N}\right) L_j^{(i-j)}\left(-\frac{\lambda y^2}{16\pi^2 N}\right) + (x \leftrightarrow y) \right]. \tag{116}$$

The large-$N$ expansion is hard [66], save for $x = y$ [57] (see section 4.1 of [60])

$$\left\langle \sum_{i,j} e^{xa_i} e^{xa_j} \right\rangle = \frac{16\pi^2 N^2}{\lambda x^2}\left[I_1\left(\frac{x\sqrt{\lambda}}{2\pi}\right)\right]^2 + \frac{x\sqrt{\lambda}}{4\pi} \tag{117}$$

$$\times \left[I_0\left(\frac{x\sqrt{\lambda}}{2\pi}\right)I_1\left(\frac{x\sqrt{\lambda}}{2\pi}\right) + \frac{1}{6}I_1\left(\frac{x\sqrt{\lambda}}{2\pi}\right)I_2\left(\frac{x\sqrt{\lambda}}{2\pi}\right)\right] + O(N^{-2}).$$

When the dust settles in (109), one obtains

$$\mathbb{W}(x,x) = \frac{x\sqrt{\lambda}}{4\pi}I_0\left(\frac{x\sqrt{\lambda}}{2\pi}\right)I_1\left(\frac{x\sqrt{\lambda}}{2\pi}\right) + O(N^{-2}). \tag{118}$$

### 4.3.2 $\mathcal{N} = 2^*$ SYM

The circular Wilson loop in $\mathcal{N} = 2^*$ SYM with hypermultiplet mass $m$ on the sphere $S^4$ of radius $R$ is measured by the matrix model $\mathbb{W}(2\pi)$ with[14]

$$\mu(a) = \frac{a^2 H^2(a)}{H(a-M)H(a+M)}, \qquad H(a) = \prod_{n=1}^{\infty}\left(1 + \frac{a^2}{n^2}\right)^n e^{-\frac{a^2}{n}}, \qquad M = mR. \tag{119}$$

The massless/small-radius limit $M \to 0$ makes the theory flow to the UV theory in section 4.3.1. The properties of $H$ are in appendix A of [67].

**Small-coupling solution.**   The integral representation[15]

$$\gamma(a) = 2 + 2a\frac{H'(a)}{H(a)} - a\frac{H'(a-M)}{H(a-M)} - a\frac{H'(a+M)}{H(a+M)} \tag{120}$$

$$= 2 + 4a\int_0^{\infty} d\omega \frac{\sin^2(M\omega)}{\sinh^2\omega}\sin(2\omega a),$$

allows systematic differentiation: we expand

$$(\sinh\omega)^{-2} = \sum_{k=1}^{\infty} 4k e^{-2k\omega}, \tag{121}$$

swap $x$-derivative and $\omega$-integration

$$\left(\frac{d}{dx}\right)^n x\sin(2\omega x)\Big|_{x=0} = (-)^{\frac{n}{2}+1}n(2\omega)^{n-1}, \tag{122}$$

then integrate and sum over $k$. The result for $n = 2, 4, \dots$ is[16]

$$\gamma^{(n)} = -2(-)^{\frac{n}{2}}n!(2\zeta_{n-1} - \zeta_{n-1,1-iM} - \zeta_{n-1,1+iM} - iM\zeta_{n,1-iM} + iM\zeta_{n,1+iM}), \tag{123}$$

in terms of Riemann and Hurwitz zeta functions

$$\zeta_s = \sum_{k=1}^{\infty} k^{-s}, \qquad \zeta_{s,a} = \sum_{k=0}^{\infty}(k+a)^{-s}. \tag{124}$$

---

[14]Instanton contributions to the measure are dropped because they should become exponentially suppressed at large $N$. There is also a divergent factor which has no incidence on expectation values and it is therefore removed.

[15]The identity holds for real argument and descends from $\mathcal{K}''(x)$ in [67].

[16]The case $n = 2$ holds under an operation of limit.

The result for $n = 0$ complies with the assumption $\gamma^{(0)} = 2$.

A prediction up to $N^{-6}$ and $\lambda^6$ follows from (96)-(99). We are able to match a matrix-model prediction [29] and an arduous two-loop calculation in gauge theory [38]

$$\mathbb{W}(2\pi) - \frac{1}{N} e^{\frac{\lambda}{8N}} L_{N-1}^{(1)}\left(-\frac{\lambda}{N}\right) = \frac{(N^2-1)\lambda^2}{64\pi^2 N^2}\left[\psi^{(0)}(1+iM) + \psi^{(0)}(1-iM) + iM\psi^{(1)}(1+iM)\right.$$
$$\left. -iM\psi^{(1)}(1-iM) + 2\gamma\right] + O(\lambda^4), \qquad (125)$$

where $\psi^{(n)}(x) = \left(\frac{d}{dx}\right)^{n+1}\Gamma(x)$ and $\gamma = -\psi^{(0)}(1)$ is the Euler's constant. Further orders can be generated with the Mathematica notebooks.

**Loop equation.** We push the solution to very high order thanks to a simpler form of the loop equation in the $\mathcal{N} = 2^*$ model

$$\mathbb{W}'(x) = \frac{\lambda}{16\pi^2}\int_0^x ds\left[\mathbb{W}(s)\mathbb{W}(x-s) + \frac{1}{N^2}\mathbb{W}(s, x-s)\right] \qquad (126)$$
$$-\frac{\lambda}{4\pi^2}\int_0^\infty d\omega \frac{\sin^2(M\omega)}{\sinh^2\omega}\,\mathrm{Im}[\mathbb{W}(2i\omega)\mathbb{W}(x-2i\omega)]$$
$$+\frac{1}{N^2}\mathbb{W}(2i\omega, x-2i\omega)\Big].$$

The proof repeats section 2.2 with the substitution of (23) and (24) with the single integral representation (120). The properties $\mathbb{W}(z^*) = (\mathbb{W}(z))^*$ and $\mathbb{W}(z^*, w^*) = (\mathbb{W}(z, w))^*$ under complex conjugation pull out the imaginary part Im. The first line is the (finite-$N$ version of the) equation (102). In the second line the single integral is due to the mass-dependent deformation of the Vandermonde measure (120) and offers an edge over the general equation (27). This could be appreciated in the linear-algebra approach of section 3.3 or numerically.[17] Here we explore a shortcut inspired by functional analysis.

The equation closes on a single function $\mathbb{W}(x)$ in the planar limit. The solution is the fixed point

$$\mathbb{W}(x) = F[\mathbb{W}](x), \qquad \mathbb{W}(0) = 1, \qquad (127)$$

of the functional

$$F[g](x) = 1 + \frac{\lambda}{16\pi^2}\int_0^x dy \int_0^y ds\, g(s)g(y-s) \qquad (128)$$
$$-\frac{\lambda}{4\pi^2}\int_0^x dy \int_0^\infty d\omega \frac{\sin^2(M\omega)}{\sinh^2\omega}\,\mathrm{Im}[g(2i\omega)g(y-2i\omega)],$$

in the space of holomorphic functions on $\mathbb{C}$. The form of the equation restricts the analysis to the strip $\mathrm{Re}(x) \in [0, 2\pi]$ as noted below (27). We construct a sequence of approximants $\{\mathbb{W}_{(n)}(x)\}_{n=0,1,...}$ by iteration $\mathbb{W}_{(n)} = F[\mathbb{W}_{(n-1)}]$. In perturbation theory we close them on polynomials seeded by the normalisation constant $\mathbb{W}_{(0)}(x) = 1$ in such a way that the fixed-point iteration converges: the approximants are partial sums

$$\mathbb{W}_{(n)}(x) = \sum_{\ell=0}^n \mathbb{W}_\ell(x)\left(\frac{\lambda}{16\pi^2}\right)^\ell, \qquad (129)$$

---

[17]Numerics at strong coupling may benefit from the further step of trading the $s$-integration with the integral kernel $\hat{R}$ of (30).

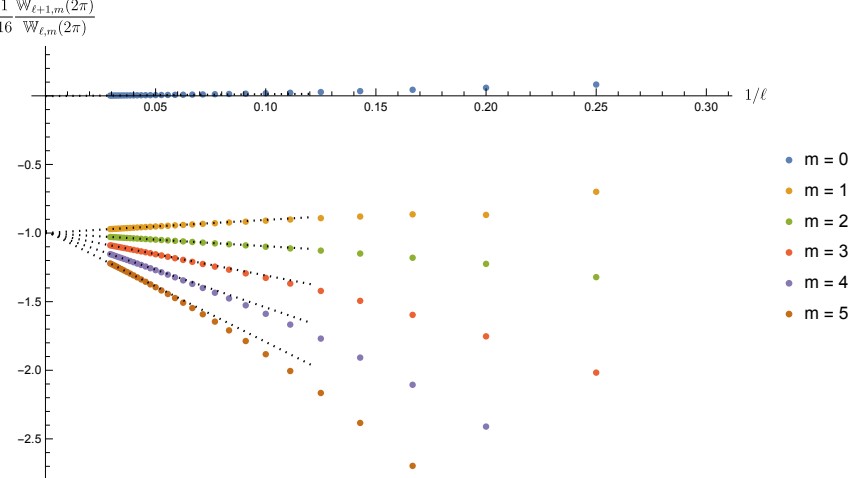

Figure 1: Domb-Sykes plot to estimate the radius of convergence of the perturbative solution at small mass. Each colours denote a set of Taylor coefficients for a $M^{2m}$-correction to the Wilson loop. The points settle on asymptotes (dotted lines) with intercepts $r_m^{-1} = \pi^2/(\lambda R_m)$. The fits of the last 5 data points give $r_2^{-1} = -0.999$, $r_2^{-1} = -1.000$, $r_3^{-1} = -0.998$, $r_4^{-1} = -0.992$ and $r_5^{-1} = -0.983$. The near-zero value $r_0^{-1} = -0.002$ detects the infinite radius of the zero-mass solution (106). The discrepancy from 0, due to a finite number of coefficients, may be taken as a rough estimate of the error on the other $r_m^{-1}$. Data sets with larger $m$ are bending downwards in the fitting window. This fact signals a slower convergence rate and can explain why the estimates of $r_4^{-1}$ and $r_5^{-1}$ are slightly above the value $-1$.

generated by

$$\mathbb{W}_\ell(x) = \int_0^x dy \int_0^y ds \sum_{\ell'=0}^{\ell-1} \mathbb{W}_{\ell'}(s)\mathbb{W}_{\ell-\ell'-1}(y-s) \tag{130}$$

$$-4\int_0^x dy \int_0^\infty d\omega \frac{\sin^2(M\omega)}{\sinh^2\omega} \sum_{\ell'=0}^{\ell-1} \mathrm{Im}\left[\mathbb{W}_{\ell'}(2i\omega)\mathbb{W}_{\ell-\ell'-1}(y-2i\omega)\right].$$

Every approximant is a polynomial of degree $\ell$ in $x^2$. The $M$-dependence shows up in a complicated manner through the derivatives (123): we perform the $\omega$-integrals as below (120)

$$\int_0^\infty d\omega \frac{\sin^2(M\omega)}{\sinh^2\omega}(2i\omega)^n = \frac{i\gamma^{(n+1)}}{4(n+1)}, \quad n = 1, 3, \ldots \tag{131}$$

The imaginary part filters out the cases with even $n$.[18]

**Small-$M$ analysis.** We estimate the radius of convergence of the perturbative solution by applying the ratio test to a finite number of its Taylor coefficients in (129). We run the algorithm above to collect those with $\ell = 0, 1, \ldots, 35$ as exact functions of $M$.

---

[18]A numerical approach to (127) and (128) could exploit the fast convergence of the exact seed (105) at zero mass to approximate solutions at small $M$. Each iteration can analytically continue an approximant on the real segment $x \in [0, 2\pi]$ to the complex strip $\mathrm{Re}(x) \in [0, 2\pi]$ by means of the Cauchy-Riemann equations and can perform the $\omega$-integral numerically. The algorithm has the unique advantage to work at finite coupling. However the fixed-point convergence is sensitive to accurate continuations far off the real axis.

At small $M$ we expand[19]

$$\mathbb{W}_\ell(x) = \sum_{m=0}^{\infty} \mathbb{W}_{\ell,m}(x) M^{2m}, \tag{132}$$

and extract the coefficient sets $m = 0, 1, \ldots, 5$. The analysis in figure 1 is compatible with the fact that the series at fixed power of $M^2$ have a common radius $|R_m|$, with

$$\frac{1}{R_m} = \frac{\lambda}{16\pi^2} \lim_{\ell \to \infty} \frac{\mathbb{W}_{\ell+1,m}(2\pi)}{\mathbb{W}_{\ell,m}(2\pi)} \approx -\frac{\lambda}{\pi^2}, \qquad m = 1, 2, \ldots, 5. \tag{133}$$

The ratio means the radius-limiting singularity lies on the negative axis at $\lambda \approx -\pi^2$. The lack of analytic control on all Taylor coefficients prevents a non-perturbative resummation. However, the estimated value agrees with the exact critical value $\lambda = -\pi^2$ that affects the free energy [32, 67], the circular Wilson loop [67] (see (135) below) and its phase transitions in the decompactification limit $M \to \infty$ [34] of $\mathcal{N} = 2^*$ SYM. Convergence-limiting singularities are ubiquitous phenomena of perturbative expansions in $\mathcal{N} = 2$ theories,[20] affecting the free energy [71], local operators and circular BPS Wilson loops [41, 42, 72–75].

We check that the truncated solution at order $M^2$

$$M^2 \sum_{\ell=0}^{35} \mathbb{W}_{\ell,1}(2\pi) \left( \frac{\lambda}{16\pi^2} \right)^\ell, \tag{134}$$

matches the perturbative expansion of the finite-coupling formula [67][21]

$$\mathbb{W}(2\pi) - \frac{2}{\sqrt{\lambda}} I_1\left(\sqrt{\lambda}\right) = 2\pi M^2 \int_0^\infty d\omega \frac{\omega J_1\left(\frac{\sqrt{\lambda}\omega}{\pi}\right)}{(\omega^2 + \pi^2) \sinh^2 \omega} \tag{135}$$
$$\times \left[ \pi I_0\left(\sqrt{\lambda}\right) J_1\left(\frac{\sqrt{\lambda}\omega}{\pi}\right) - \omega I_1\left(\sqrt{\lambda}\right) J_0\left(\frac{\sqrt{\lambda}\omega}{\pi}\right) \right] + O(M^4),$$

where $I_n$ and $J_n$ are the Bessel functions of the first kind. To this end we expand (135) for small $\lambda$ and integrate

$$\int_0^\infty d\omega \frac{\omega^n}{\sinh^2 \omega} = \frac{n!}{2^{n-1}} \zeta_n, \quad n = 2, 3, \ldots \tag{136}$$

The denominator $\omega^2 + \pi^2$ cancels order by order. The estimated leading singularity agrees with the logarithmic branch point $\lambda = -\pi^2$ that affects (135).

**Finite-$M$ analysis.** The study is hindered by the slow rate of convergence of the solution. Evidence of this trend is appreciated in figure 1. The behaviour of the coefficients, some of them in figure 2, suggests that they are positive for $\ell \leqslant \ell^*(M)$ and alternate in sign beyond a

---

[19]The small-coupling and small-mass limits commute.

[20]They are expected for a generic observable in a finite theory [8] including $\mathcal{N} = 4$ SYM. Branch-point singularities show up in the single-magnon dispersion relation [68, 69] and the modern quantum algebraic treatment [70] for dimensions of local operators. Convergence properties should be ascribed to the combinatorics of planar graphs and are typically insensitive to the particular observable. The circular Wilson loop is an exception due to massive diagram cancellations [27], for which the perturbative series has an infinite radius of convergence and it is resummed to (106). The richer graph content of the observable in $\mathcal{N} = 2^*$ SYM shifts the radius to a finite value. It may be related to integrability structures yet to be fully clarified.

[21]Following the ideas in [76], this first term in the Taylor expansion in $M^2$ can be related to the integrated two-point correlator in the $\mathcal{N} = 4$ super-Yang-Mills [77, 78].

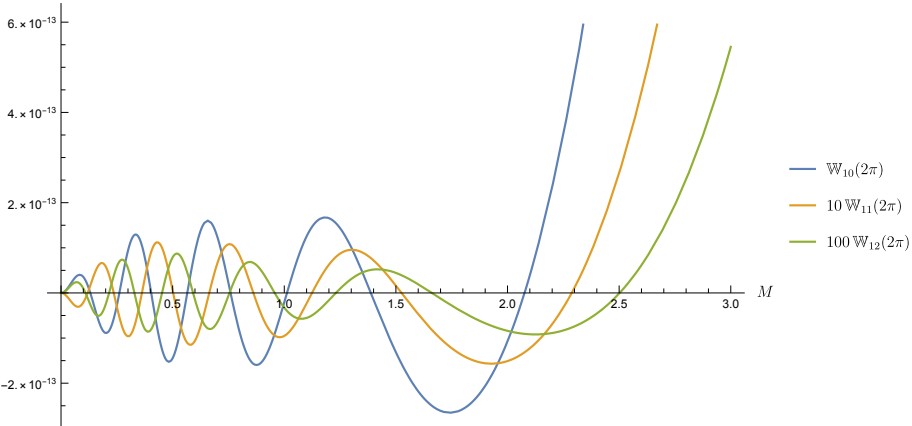

Figure 2: Analysis of the perturbative solution at finite mass. The Taylor coefficients $\mathbb{W}_\ell(2\pi)$ are positive at $M = 0$, oscillate in a finite region and grow logarithmically to positive infinity. The number of zeros seems to grow linearly in $\ell$ and their positions to spread over the positive axis. Notice two curves are magnified to fit the window.

certain order $\ell > \ell^*(M)$. The small number of extracted coefficients is insufficient to claim that there exists a tail without consecutive sign repetitions,[22] or to pinpoint a radius of convergence by root test applied to such tail. We remind that one expects the radius to be a function of $M$ on general grounds.

**Large-$M$ analysis.** The Taylor coefficients in (129) expand in $\log M$ and $M^{-1}$:

$$\mathbb{W}_0(2\pi) = 1, \qquad \mathbb{W}_1(2\pi) = 2\pi^2,$$

$$\mathbb{W}_\ell(2\pi) = 2\pi^2(4\log M)^{\ell-1} + \sum_{\ell'=0}^{\ell-2} c_{\ell'}(\log M)^{\ell'} + o(M^0), \quad \ell = 2, 3, \dots, \tag{137}$$

hence the series breaks down by the ratio test. At exponentially-large $M$ new singularities should settle in on the positive $\lambda$-axis [34],[23] although their arise is screened off by the non-commutativity of the limits $\lambda \to 0$ and $M \to \infty$.

The structure of (137) is a corollary of the measure $\gamma^{(2)} = 8\log(e^{\gamma+1}M) + O(M^{-1})$ and $\gamma^{(n)} = O(M^0)$ for $n \neq 2$, and of the solution $\mathbb{W}_\ell(x)$, which is made of products of $\gamma^{(n)}$ with the leading term $2\pi^2(\gamma^{(2)}/2)^{\ell-1}$. Physically (137) connects to the renormalisation-group properties of $\mathcal{N} = 2^*$ SYM [29, 32, 67]. In fact the infinitely-heavy matter can be integrated out, leaving $\mathcal{N} = 2$ SYM with one vector multiplet to the leading approximation. The "kinematic" scale $M$ sets the dynamically-generated scale $\Lambda R = M \exp(-4\pi^2/\lambda + \gamma + 1)$ and the coupling

$$\lambda_R = \frac{\lambda}{1 - \beta\lambda\log(e^{\gamma+1}M)}, \tag{138}$$

where $\beta = (4\pi^2)^{-1}$ is basically the numerical factor in the beta function of $\mathcal{N} = 2$ SYM [29]. If one insists on taking $\lambda \to 0^-$ and later $M \to \infty$, the Wilson loop admits an expansion similar to (129)

$$\mathbb{W}(2\pi) = \sum_{\ell=0}^{\infty} C_\ell \left(\frac{\lambda_R}{16\pi^2}\right)^\ell, \quad \text{in} \quad \mathcal{N} = 2 \text{ SYM}, \tag{139}$$

---

[22]A convergence-limiting singularity on the negative $\lambda$-axis would imply such behaviour. Only the infinite-mass limit allows the Wilson loop to undergo phase transitions on the positive axis [34].

[23]See figure 2 in [67].

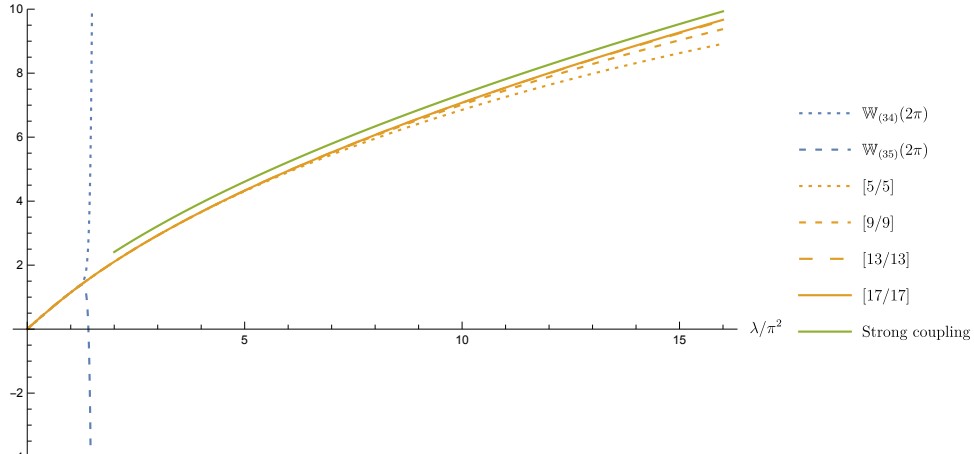

Figure 3: Padé approximation of the perturbative solution for $M = 0.5$. The polynomials (145) of degrees $n = 34$ and $n = 35$ (blue curves) are indistinguishable in the interval $\lambda \lesssim 1$ and diverge at $\lambda \approx 1$ near the radius of convergence. The blow-up with opposite signs of even- and odd-degree polynomials diagnoses a singularity on the negative $\lambda$-axis. The Padé approximants (146) (orange curves) reconstruct the polynomials up to $\lambda \lesssim 1$, extend them beyond the perturbative region and stabilise around the asymptote (142) (green curve). They become unreliable at larger $\lambda$ where they asymptote towards horizontal lines (not shown). Round-off errors in the evaluation of the polynomials can impair the precision of Padé approximants before reaching the plateaus.

with $C_0 = 1$ and $C_1 = 2\pi^2$,[24] in the renormalised coupling

$$\lambda_R = \sum_{\ell'=0}^{\infty} \lambda^{\ell'+1} \left( \beta \log(e^{\gamma+1} M) \right)^{\ell'} . \tag{140}$$

Replacing (140) into (139), the coefficient of $(\lambda/(16\pi^2))^\ell$ reads

$$C_1 \left( 16\pi^2 \beta \log M \right)^{\ell-1} , \tag{141}$$

and reproduces what observed in (137).

**Large-coupling resummation.** Finally we extrapolate the perturbative series beyond the convergence radius by means of Padé resummation [79, 80] in the spirit of [41, 72, 73, 75, 81] and compare to independent analytic results at strong coupling. This sets an important test of the solution. The Wilson loop was calculated by saddle-point techniques from the matrix model:

$$\log \mathbb{W}(2\pi) = 2\pi\mu + \log \left[ \frac{2^{3/2}}{\pi\mu^{3/2}} \left( \hat{g}(2\pi i) + \frac{1}{2^{5/2}\pi} \right) \right] + o(\lambda^0). \tag{142}$$

The leading order is that of the Gaussian model with mass-rescaled coupling [33]

$$\mu = \frac{\sqrt{\lambda(1+M^2)}}{2\pi}, \tag{143}$$

---

[24]They match those (137) in the massive theory. The first coefficient is the normalisation. The one-loop term is mass independent because in gauge theory it stems solely from the free gluon propagator.

and the $O(1)$-correction is the solution of a variant of the Wiener-Hopf problem [35]

$$\hat{g}(\omega) = \frac{i^{3/2}\sqrt{\pi}}{2\omega\sqrt{\omega+i0^+}}\left[\frac{M^2\sinh^2\frac{\omega}{2}-\sin^2\frac{M\omega}{2}}{\sinh^2\frac{\omega}{2}+\sin^2\frac{M\omega}{2}}+(M^2+1)^2\omega e^{-\frac{i\phi\omega}{2\pi}}\frac{\Gamma\left(\frac{M-i}{2\pi}\omega\right)\Gamma\left(-\frac{M+i}{2\pi}\omega\right)}{\Gamma^2\left(-\frac{i\omega}{2\pi}\right)}\right.$$
$$\left.\times\sum_{n=1}^{\infty}\frac{(-)^n}{n\,n!}\left(\frac{e^{\frac{i\phi n}{M-i}}}{\omega-\frac{2\pi n}{M-i}}\frac{\Gamma\left(\frac{M+i}{M-i}n\right)}{\Gamma^2\left(\frac{i}{M-i}n\right)}+\frac{e^{-\frac{i\phi n}{M+i}}}{\omega+\frac{2\pi n}{M+i}}\frac{\Gamma\left(\frac{M-i}{M+i}n\right)}{\Gamma^2\left(-\frac{i}{M+i}n\right)}\right)\right], \quad (144)$$
$$\phi = 2M\arctan M - \log(M^2+1).$$

The Wilson loop diverges exponentially, hence our analysis should consider the logarithm of a truncated solution (129)

$$\log\mathbb{W}_{(n)}(2\pi) = \sum_{\ell=1}^{n}c_\ell\left(\frac{\lambda}{16\pi^2}\right)^\ell + O(\lambda^{n+1}). \quad (145)$$

For example one measures $c_1 = 2\pi^2$ and $c_2 = \pi^2\gamma^{(2)} - 2\pi^4/3$ with $\gamma^{(2)}$ in (123).[25] The cutoff $n = 35$ suffices to deliver good results at moderate values $M \lesssim 1$. When the data is limited by a finite number of coefficients of (145), it is useful to define the Padé approximants[26]

$$P_{[M/K]}(\lambda) = \left[\sum_{\ell=1}^{35}c_\ell\left(\frac{\lambda}{16\pi^2}\right)^\ell\right]_{[M/K]}. \quad (146)$$

The square-root scaling and the expansion in fractional powers of (142) force a choice between $P_{[K/K]}(\lambda) = O(\lambda^0)$ and $P_{[K+1/K]}(\lambda) = O(\lambda)$. As an example, figure 3 shows that diagonal approximants are an excellent interpolation between the perturbative solution (145) at $\lambda \lesssim 1$ and the prediction (142) in a wide non-perturbative region. In particular, they better approximate (142) for increasing Padé degree $K$ at fixed $\lambda$.[27] Precise numerical results and the type of the singularity could be investigated by series accelerations and a conformal map [73].

## 4.4 Hoppe model

We also cover the model put forward and solved in the planar limit by [22], with

$$\mu(a) = \frac{a^2}{a^2+1}. \quad (147)$$

The Gaussian case $V(a) = a^2/(8\pi^2\lambda)$ was revisited via a saddle-point integral equation for the planar distribution of the matrix eigenvalues [23]. Our result is (96)-(99) with $\gamma^{(n)} = 2(-)^{\frac{n}{2}}n!$ and the values of $\tilde{c}_{n,\ell,m}$ multiplied by $(8\pi^2)^{2\ell}$ in order to compensate for $\tilde{T}_2 = (8\pi^2)^{-2} \neq 1$. The non-zero $N^0$-coefficients read up to $\lambda^6$

$$\tilde{c}_{0,1,2} = 32\pi^4, \qquad \tilde{c}_{0,2,2} = -4096\pi^8,$$
$$\tilde{c}_{0,2,4} = \frac{1024\pi^8}{3}, \qquad \tilde{c}_{0,3,2} = 1835008\pi^{12},$$

---

[25]The authors of [82] employ a trick to go from the first two terms to large coupling in the case of the cusp anomaly of the light-like Wilson cusp. Here it would find the constants $C_1(M)$ and $C_2(M)$ such that $C_1(M)\log\mathbb{W}_{(2)}(2\pi) + C_2(M)[\log\mathbb{W}_{(2)}(2\pi)]^2 = \lambda$, regard this as an *exact* algebraic equation and invert at large coupling. Note that such solution would have the same expansion in $\lambda^{-1/2}$ of the exact formula (142). However the coarse numerical agreement deteriorates when $M$ nears the zero of $C_2(M)$.

[26]The degree of (145) puts the bound $M + K \leqslant n$ on the degrees of the numerator $M$ and the denominator $K$ of the rational approximation.

[27]They eventually fall short of providing asymptotic expressions due to the wrong asymptotics, see caption of figure 3.

$$\tilde{c}_{0,3,4} = -\frac{262144\pi^{12}}{3}, \qquad \tilde{c}_{0,3,6} = \frac{16384\pi^{12}}{9}, \qquad \tilde{c}_{0,4,2} = -1157627904\pi^{16},$$

$$\tilde{c}_{0,4,4} = 46137344\pi^{16}, \qquad \tilde{c}_{0,4,6} = -\frac{2097152\pi^{16}}{3}, \qquad \tilde{c}_{0,4,8} = \frac{262144\pi^{16}}{45}, \qquad (148)$$

$$\tilde{c}_{0,5,2} = -863288426496\pi^{20}, \qquad \tilde{c}_{0,5,4} = -31675383808\pi^{20},$$

$$\tilde{c}_{0,5,6} = \frac{6308233216\pi^{20}}{15}, \qquad \tilde{c}_{0,5,8} = -\frac{134217728\pi^{20}}{45},$$

$$\tilde{c}_{0,5,10} = \frac{8388608\pi^{20}}{675}, \qquad \tilde{c}_{0,6,2} = -712483534798848\pi^{24},$$

$$\tilde{c}_{0,6,4} = 24945170055168\pi^{24}, \qquad \tilde{c}_{0,6,6} = -309237645312\pi^{24},$$

$$\tilde{c}_{0,6,8} = \frac{300064771072\pi^{24}}{15}, \qquad \tilde{c}_{0,6,10} = -\frac{1073741824\pi^{24}}{135},$$

$$\tilde{c}_{0,6,12} = \frac{268435456\pi^{24}}{14175},$$

the $N^{-2}$-coefficients

$$\tilde{c}_{1,2,2} = 4096\pi^8, \qquad \tilde{c}_{1,2,4} = \frac{512\pi^8}{3}, \qquad \tilde{c}_{1,3,2} = -1835008\pi^{12},$$

$$\tilde{c}_{1,3,4} = \frac{262144\pi^{12}}{3}, \qquad \tilde{c}_{1,3,6} = \frac{32768\pi^{12}}{9}, \qquad \tilde{c}_{1,4,2} = 1157627904\pi^{16},$$

$$\tilde{c}_{1,4,4} = -46137344\pi^{16}, \qquad \tilde{c}_{1,4,8} = \frac{262144\pi^{16}}{9}, \qquad \tilde{c}_{1,5,2} = -695784701952\pi^{20}, \quad (149)$$

$$\tilde{c}_{1,5,4} = 27380416512\pi^{20}, \qquad \tilde{c}_{1,5,8} = -\frac{67108864\pi^{20}}{9},$$

$$\tilde{c}_{1,5,10} = \frac{16777216\pi^{20}}{135}, \qquad \tilde{c}_{1,6,2} = 241617680203776\pi^{24},$$

$$\tilde{c}_{1,6,4} = -12094627905536\pi^{24}, \qquad \tilde{c}_{1,6,6} = -\frac{300647710720\pi^{24}}{3},$$

$$\tilde{c}_{1,6,8} = \frac{15032385536\pi^{24}}{3}, \qquad \tilde{c}_{1,6,10} = -\frac{7516192768\pi^{24}}{135},$$

$$\tilde{c}_{1,6,12} = \frac{134217728\pi^{24}}{405},$$

the $N^{-4}$-coefficients

$$\tilde{c}_{2,4,6} = \frac{2097152\pi^{16}}{3}, \qquad \tilde{c}_{2,4,8} = \frac{131072\pi^{16}}{15}, \qquad \tilde{c}_{2,5,2} = -167503724544\pi^{20},$$

$$\tilde{c}_{2,5,4} = 4294967296\pi^{20}, \qquad \tilde{c}_{2,5,6} = -\frac{6308233216\pi^{20}}{15},$$

$$\tilde{c}_{2,5,8} = \frac{469762048\pi^{20}}{45}, \qquad \tilde{c}_{2,5,10} = \frac{96468992\pi^{20}}{675},$$

$$\tilde{c}_{2,6,2} = 470865854595072\pi^{24}, \qquad \tilde{c}_{2,6,4} = -12850542149632\pi^{24},$$

$$\tilde{c}_{2,6,6} = \frac{1228360646656\pi^{24}}{3}, \qquad \tilde{c}_{2,6,8} = -\frac{105226698752\pi^{24}}{15},$$

$$\tilde{c}_{2,6,10} = \frac{3758096384\pi^{24}}{135}, \qquad \tilde{c}_{2,6,12} = \frac{1879048192\pi^{24}}{2025},$$

and the $N^{-6}$-coefficients

$$\tilde{c}_{3,6,10} = \frac{536870912\pi^{24}}{15}, \qquad \tilde{c}_{3,6,12} = \frac{67108864\pi^{24}}{315}. \qquad (150)$$

# 5 Strong coupling: Solution

We now turn to strong coupling. In what follows we develop a method to systematically solve the loop equation in that regime. This is particularly important for localisation matrix models which often have dual holographic description that becomes simple at strong coupling.

To discuss the strong-coupling limit we redefine the potential as

$$V(a) = \hbar^{2\nu-2} U(\hbar a). \tag{151}$$

If the potential is quadratic and $\nu = 1$, the new parameter $\hbar$ is literally the inverse of the coupling. Indeed, for $V(a) = a^2/2g^2$ the rescaled potential is $U(x) = x^2/2$, and $\hbar = 1/g$. The anharmonic terms, if present, must follow a particular pattern in the $\hbar \to 0$ limit. In the general parameterisation (5), the individual couplings $T_n$ scale as $\hbar^n$, while the parameter $\nu$ defines the overall scaling weight of the potential. We show later that this parameter is not arbitrary and is dictated by the behaviour of the integration measure $\mu(a)$ at large $a$, in order for the strong-coupling limit to be well-defined.

The strong-coupling regime corresponds to $\hbar \to 0$. When the potential is quadratic, expanding in $\hbar$ is equivalent to expanding in $1/g$, exactly opposite to the weak-coupling expansion considered so far. For example, in the $\mathcal{N} = 2^*$ SYM theory or any other model relevant for AdS/CFT, $V(a) = 8\pi^2 a^2/\lambda$ and thus $\hbar = 4\pi/\sqrt{\lambda}$. This coupling coincides with the natural expansion parameter in the dual string theory, making the strong-coupling expansion in the matrix model equivalent to the weak-coupling expansion on the string worldsheet.

We develop a systematic procedure to generate expansion of the Wilson loop in $\hbar$ for the potential of the form (151) and an arbitrary measure. Our motivation comes from localisation and string theory, but the method is completely general and we expect it to have other applications.

The form of the potential (151) suggests to rescale the integration variables $a_i \to a_i/\hbar$. This tacitly assumes that typical $a_i$'s in the eigenvalue integral are very large, of order $1/\hbar$. The measure $\mu(a)$, it may seem, can be replaced by its large-argument asymptotics and then expanded in $1/a$ to generate the next orders of the expansion. The loop equations, as we shall see, do justify this simple reasoning up to a small but important caveat. The measure depends on the eigenvalue difference and the typical distance between nearby eigenvalues may be of order one or smaller even if all eigenvalues are large. This is especially true at large $N$ when the eigenvalues form a continuous distribution and can be found arbitrary close to one another. More care is thus needed, and our goal is to set up a systematic procedure to develop the strong-coupling expansion that would take this effect into account.

For the Wilson loop, the two regimes have to be distinguished at strong coupling: short loops $\mathbb{W}(x)$ with $x \sim \hbar$ and long loops with $x \sim 1$. The naive strong-coupling approximation holds for short loops, while for long loops (more interesting for applications) nearby eigenvalues start to become important and the answer depends on the measure in a non-trivial way. The two regimes match in the overlapping region of validity $1 \gg x \gg \hbar$.

## 5.1 Short loops

The natural variable for short loops is $x/\hbar$. The integration variables in the loop equation, be it (27) or (30), need to be rescaled accordingly. The kernels can then be replaced by their small-argument asymptotics (in the Fourier space) or the large-argument ones in the original $a$-variables. To proceed further we need to know how the measure, equivalently the kernel in the loop equation, behaves in this limit. We will assume a power-like asymptotics (cf. (31)), which covers a large class of physically interesting eigenvalue models:

$$\hat{R}(\omega) \stackrel{\omega \to 0}{\simeq} -\pi\beta \operatorname{sign} \omega |\omega|^{2\nu-2} \equiv \hat{R}_\infty(\omega). \tag{152}$$

All eigenvalue integrals with measure growing not faster than polynomially fall into this class. Indeed, for

$$\mu(a) \overset{a\to\infty}{\simeq} C\,a^{2\beta}\,, \tag{153}$$

and $\beta > 0$,

$$\hat{\gamma}(\omega) \overset{\omega\to 0}{\simeq} 4\pi\beta\,\delta(\omega)\,, \qquad \hat{R}(\omega) \overset{\omega\to 0}{=} -\pi\beta\,\text{sign}\,\omega\,, \tag{154}$$

which in our notations corresponds to $\nu = 1$. This behavior is found for example in the $\mathcal{N} = 2^*$ localisation integral. In the Hoppe model the kernel has a power-like asymptotics with an exponent different from one. We will discuss the Hoppe model in greater detail later.

Denoting by $W_\infty$ the leading-order strong-coupling approximation for short loops, we arrive at the equation

$$\frac{i\hbar^{2\nu-1}}{2}\,U'\left(-i\hbar\frac{\partial}{\partial\kappa}\right)W_\infty(\kappa) = \int\limits_{-\infty}^{+\infty}\frac{d\omega}{2\pi}\,\hat{R}_\infty(\omega)W_\infty(\kappa-\omega)W_\infty(\omega)\,, \tag{155}$$

in which all the $\hbar$-dependence can be absorbed into rescaling $\kappa \to \hbar\kappa$. Notice that for this to happen the exponents in the kernel (152) and in the potential (151) must agree with one another. Upon the rescaling we get the loop equation for the $\nu$-model from sec. 2.3.

The solution (analytically continued to complex $\kappa$) is

$$\mathbb{W}_\infty(x) = \sum_{n\geqslant 0} A_n\,\frac{I_{\nu+n}\left(\frac{\mu x}{\hbar}\right)}{\left(\frac{\mu x}{\hbar}\right)^{\nu+n}}\,, \tag{156}$$

with

$$A_n = \frac{\mu^{2\nu-1}(2n+1)!}{4\pi\beta\,\Gamma(2\nu-1)}\int\limits_{-1}^{1} da\,(1-a^2)^{\nu-n-\frac{3}{2}}\mathcal{D}_n^{(\nu)}(a)\left(a\frac{\partial}{\partial a}+2\nu-1\right)U'(\mu a)\,, \tag{157}$$

which differs from (49) by an extra factor of $\beta$ from the kernel in (154). The parameter $\mu$ is to be fixed by normalisation, which we will discuss later, after taking into account the next correction in $\hbar$.

## 5.2 Long loops

We can get an insight into the behaviour of long loops by considering a short loop with $x \gg \hbar$, but $x \ll 1$. Approximations used in deriving (156) then still apply, and we can just take the $x \to \infty$ limit of that expression. All the Bessel functions have the same exponential asymptotics, taking into account the power-like prefactor we can see that the term with $n = 0$ is the largest one, and we get:

$$\mathbb{W}_\infty(x) \overset{1\gg x\gg\hbar}{\simeq} \frac{A_0}{\sqrt{2\pi}}\left(\frac{\hbar}{\mu x}\right)^{\nu+\frac{1}{2}}e^{\frac{\mu x}{\hbar}}\,. \tag{158}$$

The constant $A_0$, given by (157), can be simplified taking into account the explicit form of the first Gegenbauer polynomial (37) and integrating by parts:

$$A_0 = \frac{\mu^2}{2^\nu\sqrt{\pi}\,\beta\,\Gamma\left(\nu-\frac{1}{2}\right)}\int\limits_{-\mu}^{\mu} da\,\left(\mu^2-a^2\right)^{\nu-\frac{3}{2}}U''(a)\,. \tag{159}$$

For yet larger $x$ of order one the short-loop approximation can no longer be trusted, but it is natural to expect that exponential dependence on $x$ will persist. This suggests an Ansatz:

$$\mathbb{W}(x) = \frac{A_0 \hbar^{\nu+\frac{1}{2}}}{\sqrt{2\pi}\, \mu^{\nu+\frac{1}{2}}} \, e^{\frac{\mu x}{\hbar}} \, \mathbb{f}(x), \tag{160}$$

with some unknown function $\mathbb{f}(x)$. Matching to the short-loop solution we find:

$$\mathbb{f}(x) \overset{x \to 0}{\simeq} x^{-\nu-\frac{1}{2}}, \tag{161}$$

which will serve as a boundary condition for the equation we are going to derive.

Accordingly, the oscillating Wilson loop (29) behaves as

$$W(\kappa) = \frac{A_0 \hbar^{\nu+\frac{1}{2}}}{\sqrt{2\pi}\, \mu^{\nu+\frac{1}{2}}} \, e^{\frac{i\mu\kappa}{\hbar}} \, f(\kappa), \tag{162}$$

where $f(\kappa)$ and $\mathbb{f}(x)$ are related by analytic continuation $\kappa \to -ix$. An important remark is in order here. For real $\kappa$ the Wilson loop no longer grows but oscillates and this gives rise to subtleties with the analytic continuation. An instructive example is the Bessel function. The modified Bessel function grows exponentially:

$$I_\nu(x) \overset{x \to \infty}{\simeq} \frac{1}{\sqrt{2\pi x}} \, e^x, \tag{163}$$

while its analytic continuation, the ordinary Bessel function asymptotes to

$$J_\nu(\kappa) \overset{\kappa \to \infty}{\simeq} \sqrt{\frac{2}{\pi\kappa}} \cos\left(\kappa - \frac{\pi\nu}{2} - \frac{\pi}{4}\right), \tag{164}$$

and contains oscillating exponentials of both signs, so the analytic continuation and the limit of the argument going to infinity do not commute. To suppress the exponential with the wrong sign, and to continue using the single exponent in (162), we need to give $\kappa$ a (small) negative imaginary part.

This example shows that an analytic continuation of $f(\kappa)$ is only well-defined in the lower half-plane of $\kappa$, where it coincides with $\mathbb{f}(x)$:

$$f(-ix) = \mathbb{f}(x), \qquad \text{Re}\, x > 0. \tag{165}$$

The function $f(\kappa)$ can be analytically extended to the real line and to the upper half-plane, but it need not coincide there with the $\hbar \to 0$ limit of the exact Wilson loop. The latter is well-defined for any value of the argument, while an analytic continuation of $f(\kappa)$ into the upper half-plane may hit singularities of various kinds.

One singularity is actually prescribed by the boundary condition (161), according to which $f(\kappa)$ has a branch point at zero, shifted slightly into the upper half-plane:

$$f(\kappa) \overset{\kappa \to 0}{\simeq} \frac{i^{-\nu-\frac{1}{2}}}{(\kappa - i\epsilon)^{\nu+\frac{1}{2}}} \equiv f_\infty(\kappa). \tag{166}$$

The branch of $(\kappa - i\epsilon)^{\nu+\frac{1}{2}}$ is defined with a cut along the positive imaginary semi-axis. Other singularities may hide further away from the origin.

Keeping this in mind, we may substitute (162) into the loop equation (30). As emphasised early on [2], an exponential Ansatz always goes through. Indeed, the exponential factors re-combine inside the integral and cancel between the two sides so long as the exponent is

linear in $\kappa$. Another simplification occurs because the exponent is large at $\hbar \to 0$, bearing certain similarity to the semi-classical wavefunction in quantum mechanics. The action of the differential operator $V'(-i\partial/\partial\kappa) = \hbar^{2\nu-1} U'(-i\hbar\,\partial/\partial\kappa)$ on the Wilson loop can then be replaced with multiplication by a constant $\hbar^{2\nu-1} U'(\mu)$, to the leading order in $\hbar$.

At a first sight this leads to problems. The power counting does not seem to match. The left-hand side is proportional to $\hbar^{2\nu-1} \times \hbar^{\nu+\frac{1}{2}} = \hbar^{3\nu-\frac{1}{2}}$, while the right-hand side scales as $\hbar^{\nu+\frac{1}{2}} \times \hbar^{\nu+\frac{1}{2}} = \hbar^{2\nu+1}$. Next, the integral over $\omega$ diverges at zero for $\nu \leqslant 3/2$ as a consequence of (166).

The two problems are not unrelated. The integral includes not only $\omega \sim 1$, but also $\omega \sim \hbar$ where the Ansatz (166) is not applicable. The short-loop contribution is proportional to $\hbar^{2\nu-1} \times \hbar^{\nu+\frac{1}{2}} = \hbar^{3\nu-\frac{1}{2}}$, a factor of $\hbar^{\frac{3}{2}-\nu}$ bigger than the contribution of long loops (proportional to $\hbar^{2\nu+1}$). To extract the short-loop contribution we can use the solution (156) inside the integral for the small range of $\omega \sim \hbar$, but first it is necessary to isolate this leading term in the equation.

This can be done by the following trick. In the dangerous region of small $\omega$ the Wilson loop is well approximated by $W_\infty$ and the kernel by $\hat{R}_\infty(\omega)$. Adding and subtracting those splits the integral into two parts:

$$
\frac{i\hbar^{2\nu-1}}{2} U'(\mu) W(\kappa) = \int_{-\infty}^{+\infty} \frac{d\omega}{2\pi} \left[ \left( \hat{R}(\omega) - \hat{R}_\infty(\omega) \right) W(\kappa-\omega) W(\omega) \right.
$$
$$
\left. + \hat{R}_\infty(\omega) W(\kappa-\omega) \left( W(\omega) - W_\infty(\omega) \right) \right]
$$
$$
+ \int_{-\infty}^{+\infty} \frac{d\omega}{2\pi} W(\kappa-\omega) \hat{R}_\infty(\omega) W_\infty(\omega) . \tag{167}
$$

In the last term the main contribution comes from $\omega \sim \hbar$ and $W(\kappa-\omega)$ there can be replaced by $W(\kappa)$ (for $\kappa \sim 1$). The remainder, one can show, integrates to[28]

$$
\int_{-\infty}^{+\infty} \frac{d\omega}{2\pi} \hat{R}_\infty(\omega) W_\infty(\omega) = \frac{i\hbar^{2\nu-1}}{2} U'(\mu) , \tag{168}
$$

and cancels the left-hand side.

The integral in the middle of (167) converges at $\omega = 0$ and thus receives no contribution from short loops. The exact Wilson loop can now be replaced by its strong-coupling asymptotics (162):

$$
\int_{-\infty}^{+\infty} \frac{d\omega}{2\pi} f(\kappa-\omega) \left( \hat{R}(\omega) f(\omega) - \hat{R}_\infty(\omega) f_\infty(\omega) \right) = 0 . \tag{169}
$$

This non-linear integral equation on the scaling function $f(\kappa)$ describes the strong-coupling behaviour of the Wilson loop for $\kappa \sim 1$.

Our approach to solving this equation is inspired by the Wiener-Hopf method, in the form it was applied to matrix models in [35, 40, 83, 84]. Our results are actually equivalent to the Wiener-Hopf solution of $\mathcal{N} = 2^*$ super-Yang-Mills [35], given by (144), even if our derivation is different and also applicable to any generalised eigenvalue model.

---

[28]This formula can be proven by sending $\kappa \to (1-i\epsilon)\infty$ in the equation (155), using (158) and replacing $V'(-i\hbar\partial/\partial\kappa)$ with $\hbar^{2\nu-1} U'(\mu)$. It can be also derived by direct integration of the exact solution (156).

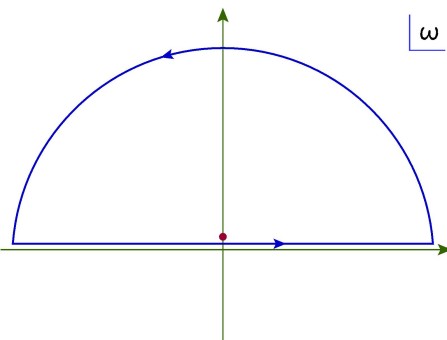

Figure 4: Closing the contour of integration in the upper half-plane.

The key point is the Riemann-Hilbert decomposition of the kernel:

$$\hat{R}(\omega) = \frac{G_+(\omega)}{G_-(\omega)}, \tag{170}$$

where $G_+(\omega)$ admits an analytic continuation into the upper half-plane and $G_-(\omega)$ into the lower one. This representation exists under very general assumptions and is unique up to simultaneous rescaling of $G_+$ and $G_-$ by the same constant.

The equation (169), as it stands, is defined on the real line. The idea is to deform the contour of integration into the complex plane, more precisely into its upper half where $f(\kappa-\omega)$ is a well defined function of $\omega$ for a fixed real $\kappa$. The analytic form of the asymptotic kernel is

$$\hat{R}_\infty(\omega) = -\pi\beta(\omega + i\epsilon)^{\nu - \frac{3}{2}}(\omega - i\epsilon)^{\nu - \frac{1}{2}}, \tag{171}$$

where the two factors have cuts along the imaginary axis in the opposite directions. The complex function so defined indeed coincides with the real-valued function (152) once the argument is restricted to the positive half of the real line, while in the complex plane it has two cuts along the positive and negative imaginary semi-axes. To extend this function to negative semi-axis we need to pass between the two branch points at $\omega = \pm i\epsilon$. The phases of the two factors rotate in the opposite direction, because their cuts extend in the opposite half-planes, producing an overall minus sign. This is the origin of the sign $\omega$ factor in (152).

The Wiener-Hopf decomposition of the asymptotic kernel, obvious from (171), defines the boundary conditions for the exact Wiener-Hopf factors:

$$G_+(\omega) \overset{\omega \to 0}{\simeq} -\pi\beta(\omega + i\epsilon)^{\nu - \frac{3}{2}}, \tag{172}$$

$$G_-(\omega) \overset{\omega \to 0}{\simeq} (\omega - i\epsilon)^{\frac{1}{2} - \nu}. \tag{173}$$

Common normalisation here is a matter of convention. Being made, it fixes the rescaling ambiguity of the Riemann-Hilbert decomposition.

With these data at hand we can proceed with the analytic continuation of (169). Taking $f_\infty$ from (166) and denoting $f/G_- = P$, we can write the integral equation as

$$\int\limits_{-\infty}^{+\infty} \frac{d\omega}{2\pi} f(\kappa-\omega)\left[ G_+(\omega)P(\omega) + \frac{\pi\beta i^{-\nu-\frac{1}{2}}(\omega + i\epsilon)^{\nu - \frac{3}{2}}}{\omega - i\epsilon} \right] = 0, \tag{174}$$

with $P$ being the new unknown function. Importantly, this function is analytic in the lower half-plane.

The next step is a contour deformation trick. If the contour shown in figure 4 encircles no singularities of the integrand, the integral will automatically be zero, and we will be done.

To achieve this we first notice that the function $f(\kappa - \omega)$ is analytic in the upper half-plane of $\omega$, and so is $G_+(\omega)$. There are two possible obstructions to closing the contour. One is the pole at $\omega = i\epsilon$. Another is possible singularities of $P(\omega)$ in the upper half-plane. We are in principle agnostic about the latter, however, any singularity of $P$ that is not localised at $\omega = i\varepsilon$ cannot be cancelled by the second term. Barring no accidental cancellations occur, such singularities better not be there and $P(\omega)$ better have a pole at $\omega = i\epsilon$ with no other singularities whatsoever. The problem reduces to matching the residues such that the pole cancels in the sum. The integrand in (174) is then analytic in the upper half-plane, the contour can be closed and the integral will be zero as requested.

Taking into account (172), the residues will match provided

$$P(\omega) = \frac{i^{-\nu-\frac{1}{2}}}{\omega - i\epsilon}\,. \tag{175}$$

We thus find:

$$f(\kappa) = \frac{i^{-\nu-\frac{1}{2}} G_-(\kappa)}{\kappa - i\epsilon}\,. \tag{176}$$

This function is manifestly analytic in the lower half-plane, has the right asymptotics (166) in virtue of (173) and solves (169) by the above contour-deformation argument.

For the Wilson loop, as defined in (160) and (162), we get:

$$\mathbb{W}(x) = \frac{i^{\frac{1}{2}-\nu} A_0 \hbar^{\nu+\frac{1}{2}}}{\sqrt{2\pi}\, \mu^{\nu+\frac{1}{2}} x}\, e^{\frac{\mu x}{\hbar}} G_-(-ix)\,. \tag{177}$$

This is our final result. The constant $A_0$ is given by (159) and $G_-$ is determined by the Riemann-Hilbert factorisation of the kernel. To close the circuit we need to find $\mu$, which can be done by imposing the normalisation condition $\mathbb{W}(0) = 1$. The asymptotic solution (177) is unsuitable for this purpose, since it applies to long loops with $x \gg \hbar$ and cannot be continued down to $x = 0$. We need to use short loops, but the solution (156) is not sufficient. Since $\mu$ enters (177) in the ratio $\mu/\hbar$ it has to be known to $\mathcal{O}(\hbar)$. Hence we need to know the short-loop solution to the next order in the strong-coupling expansion. Fortunately, the first-order corrections are simple and can be easily taken into account.

### 5.2.1 Shift of the spectral endpoint

To find the short-loop solution (156) we replaced the kernel by its asymptotic form (152) and rescaled $\omega \to \hbar\omega$. Since $\hat{R}(\omega)$ is an odd function, see (16), at small $\omega$,

$$\hat{R}(\omega) = \hat{R}_\infty(\omega)\big(1 + \mathcal{O}(\omega^2)\big)\,. \tag{178}$$

The correction term is of relative order $\omega^2 \to \hbar^2 \omega^2$, and can be neglected as long as we are interested in effects linear in $\hbar$.

At the same time, the Riemann-Hilbert factors $G_\pm(\omega)$ are neither even nor odd and, in general,

$$G_-(\omega) \stackrel{\omega \to 0}{\simeq} (\omega - i\epsilon)^{\frac{1}{2}-\nu}(1 + i\gamma\omega + \ldots)\,. \tag{179}$$

The linear term then affects the solution at $\mathcal{O}(\hbar)$.

The solution for long loops has to match with (158) at small $x$. Taking into account the first correction, that is, expanding the already known solution (177) in $x$, we find:

$$\mathbb{W}_\infty(x) \stackrel{1 \gg x \gg \hbar}{\simeq} \frac{A_0}{\sqrt{2\pi}} \left(\frac{\hbar}{\mu x}\right)^{\nu+\frac{1}{2}} e^{\frac{\mu x}{\hbar}} (1 + \gamma x + \ldots)\,. \tag{180}$$

Matching it to the general expression for short loops (156) requires to include the $n = -1$ term, absent in the zeroth-order solution, with the coefficient

$$A_{-1} = \frac{\hbar\gamma}{\mu} A_0 \,. \tag{181}$$

The solution for short loops accurate through $\mathcal{O}(\hbar)$ is thus

$$\mathbb{W}_\infty(x) = \sum_{n\geqslant 0} A_n \frac{I_{\nu+n}\left(\frac{\mu x}{\hbar}\right)}{\left(\frac{\mu x}{\hbar}\right)^{\nu+n}} + \frac{\hbar\gamma}{\mu} A_0 \frac{I_{\nu-1}\left(\frac{\mu x}{\hbar}\right)}{\left(\frac{\mu x}{\hbar}\right)^{\nu-1}} \,. \tag{182}$$

It may seem odd that the equation for short loops has not changed at $\mathcal{O}(\hbar)$ but the solution does receive an order $\hbar$ correction. There is no contradiction, however. The reason is that the key operatorial identity (39) simply gives zero for $n < 0$, according to (35). Any term with $n < 0$ can thus be added without affecting the equation. They cannot arise at the zeroth order, as they compromise the boundary conditions at $x \to \infty$, but at higher orders they are allowed and do occur by the matching argument above.

The normalisation condition $\mathbb{W}(0) = 1$ results in

$$\sum_{n\geqslant 0} \frac{A_n}{2^{\nu+n}\Gamma(\nu+n+1)} + \hbar \frac{\gamma A_0}{2^{\nu-1}\mu\Gamma(\nu)} = 1 \,, \tag{183}$$

with an extra correction term compared to (51). This equation determines $\mu$ to the $\mathcal{O}(\hbar)$ accuracy and thus completes the solution for the Wilson loop at the two first orders in the strong-coupling expansion.

To conclude, the solution for the Wilson loop is given by (177) where $\mu$ si determined by (183) and the constants $A_n$ are taken from (157).

For the Gaussian model with $U(a) = a^2/2$,

$$A_0 = \frac{\mu^{2\nu}}{2^\nu\Gamma(\nu)\beta} \,, \tag{184}$$

all other $A_n = 0$ and

$$\mu = 2\left(\Gamma(\nu)\Gamma(\nu+1)\beta\right)^{\frac{1}{2\nu}} - \hbar\gamma \,, \tag{185}$$

to the first order in $\hbar$. We checked that these equations along with (177) reproduce the first two orders of the strong-coupling expansion in the $\mathcal{N} = 2^*$ SYM theory [35]. To further illustrate our method we consider the strong-coupling expansion in the Hoppe model.

## 6 Strong coupling: Hoppe model

The Hoppe model is defined by the Gaussian potential $V(a) = a^2/2g^2$ and the measure[29]

$$\mu(a) = \frac{a^2}{a^2 + 1} \,. \tag{186}$$

The kernel of the loop equation is determined by the log-derivative of the measure.

$$R(a) = \frac{1}{a} - \frac{a}{a^2 + 1} \,, \tag{187}$$

---

[29]A more general measure $\mu(a) = a^2/(a^2 + m^2)$ does not introduce new parameters, as $m$ can be scaled away by the change of variables $a \to am$ with subsequent redefinition of the coupling.

which gives, upon Fourier transform:

$$\hat{R}(\omega) = -2\pi\, e^{-\frac{|\omega|}{2}} \sinh\frac{\omega}{2}\,. \tag{188}$$

In contradistinction to (154), the kernel vanishes at $\omega = 0$:

$$\hat{R}(\omega) \overset{\omega\to 0}{\simeq} -\pi\omega \equiv \hat{R}_\infty(\omega)\,, \tag{189}$$

and has a non-trivial index and a slope

$$\nu = \frac{3}{2}\,, \qquad \beta = 1\,. \tag{190}$$

The same index controls the scaling of the potential and thus defines the parameter of the strong-coupling expansion. Comparing the potential $V(a) = a^2/2g^2$ to the general scaling form (151) we find:

$$\hbar = g^{-2/3}\,. \tag{191}$$

We thus come to a conclusion, not completely obvious from the model's definition, that the strong-coupling expansion goes in powers of $g^{-2/3}$.

## 6.1 Short loops

For short Wilson loop we can use the solution of the $\nu$-model with the asymptotic kernel, whose explicit form for the Gaussian potential is given by (156) with the constants $A_0$ and $\mu$ in (184), (185). Ignoring, for the moment, the $\mathcal{O}(\hbar)$ shift of the endpoint, we get:

$$\mu = (3\pi)^{\frac{1}{3}}\,, \qquad A_0 = 3\sqrt{\frac{\pi}{2}}\,, \tag{192}$$

and

$$\mathbb{W}_\infty(x) = \sqrt{\frac{3}{2x^2}}\, I_{\frac{3}{2}}\left((3\pi g^2)^{\frac{1}{3}}x\right) = \frac{3}{y^3}(y\cosh y - \sinh y)\Big|_{y=(3\pi g^2)^{\frac{1}{3}}x}\,. \tag{193}$$

This corresponds to the eigenvalue density

$$\rho(a) = \frac{3}{4B^3}(B^2 - a^2)\,, \qquad B = (3\pi g^2)^{\frac{1}{3}}\,, \tag{194}$$

defined on the interval from $-B$ to $B$. The density can be found from (53), with $A_0$ taken from (192) and $\mu$ replaced by $B$ to account for the rescaling $\mu \to \mu/\hbar = \mu g^{2/3} = B$ in (156). These results are valid for $x \sim g^{-2/3}$. For longer loops the formulas from section 5.2 should be used instead.

## 6.2 Long loops

The kernel (188) can be readily factorised with the help of an analytic representation of $|\omega|$:

$$|\omega| = \frac{i\omega}{\pi}\left(\log\frac{\omega - i\epsilon}{2\pi} - \log\frac{\omega + i\epsilon}{2\pi}\right) + \omega\,, \tag{195}$$

along with representation of sinh as a product of two gamma-functions. That gives:

$$G_+(\omega) = -\frac{\pi}{\Gamma\left(1 - \frac{i\omega}{2\pi}\right)}\, e^{-\frac{\omega}{4} - \frac{i\omega}{2\pi}\left(\log\frac{\omega + i\epsilon}{2\pi} - 1\right)}\,,$$

$$G_-(\omega) = \frac{1}{\omega - i\epsilon}\,\Gamma\left(1 + \frac{i\omega}{2\pi}\right) e^{\frac{\omega}{4} - \frac{i\omega}{2\pi}\left(\log\frac{\omega - i\epsilon}{2\pi} - 1\right)}\,. \tag{196}$$

The solution derived in section 5.2 is expressed in terms of $G_-$, and in addition contains an $\mathcal{O}(\hbar)$ shift of $\mu$. The latter is determined by a constant that appears in (179), the Taylor expansion of $G_-(\omega)$ at small $\omega$. In the Hoppe model that expansion is contaminated by logarithms:

$$G_-(\omega) \overset{\omega \to 0}{\simeq} \frac{1}{\omega - i\epsilon}\left(1 - \frac{i\omega}{2\pi}\log\omega + \dots\right). \tag{197}$$

This does not have exactly the same form as (179), but if we recall that the small-$\omega$ expansion is matched to the short-loop solution at $\omega \sim \hbar$, the log term is essentially equivalent to setting

$$\gamma = -\frac{1}{2\pi}\log\hbar. \tag{198}$$

This determines the endpoint shift with the logarithmic accuracy:[30]

$$\mu = (3\pi)^{\frac{1}{3}} + \frac{1}{2\pi}\hbar\log\hbar. \tag{199}$$

The next, $\mathcal{O}(\hbar)$ term cannot be determined by this simple reasoning.

Taking this all into account we find from (177):

$$\mathbb{W}(x) = \left(\frac{3}{8\pi^2 g^4 x^6}\right)^{\frac{1}{3}}\Gamma\left(1 + \frac{x}{2\pi}\right)e^{\left[(3\pi g^2)^{\frac{1}{3}} - \frac{1}{6\pi}\log(Cg^2 x^3)\right]x}. \tag{200}$$

The numerical constant $C$ cannot be determined within this scheme. Finding it requires a more refined analysis which goes beyond the leading-log approximation.

It is no more difficult to repeat the same analysis for an arbitrary even potential, not necessarily Gaussian. We concentrated on the simplest quadratic case because the Gaussian model is exactly solvable and the results can be directly confronted with the strong-coupling asymptotics of the exact solution.

### 6.3 Comparison to exact solution

The exact solution of the Hoppe model is formulated in terms of a resolvent function $G(z)$ with the following properties:[31]

(i) The resolvent is an even function: $G(-z) = G(z)$.

(ii) It is analytic in the complex plane with two cuts $(-B \pm i/2, B \pm i/2)$ (figure 5).

(iii) The function is real and monotonous along the contour $C$ shown in the figure.

(iv) The discontinuity of the resolvent across the cut determines the eigenvalue density:

$$2\pi\rho(a) = G\left(a + \frac{i}{2} - i\varepsilon\right) - G\left(a + \frac{i}{2} + i\varepsilon\right). \tag{201}$$

(v) At infinity the resolvent behaves as

$$G(z) \overset{z \to \infty}{\simeq} \frac{z^2}{2g^2}. \tag{202}$$

---

[30]The expansion of the full kernel (188) also does not match the generic form (178), because it starts with a linear term proportional to $|\omega|$ and not quadratic in $\omega$ as had been assumed in the derivation. This entails extra corrections to short loops beyond matching, but those corrections are of order $\mathcal{O}(\hbar)$ and in the leading-log approximation can be ignored.

[31]A derivation can be found in [22, 23]; we just quote the results.

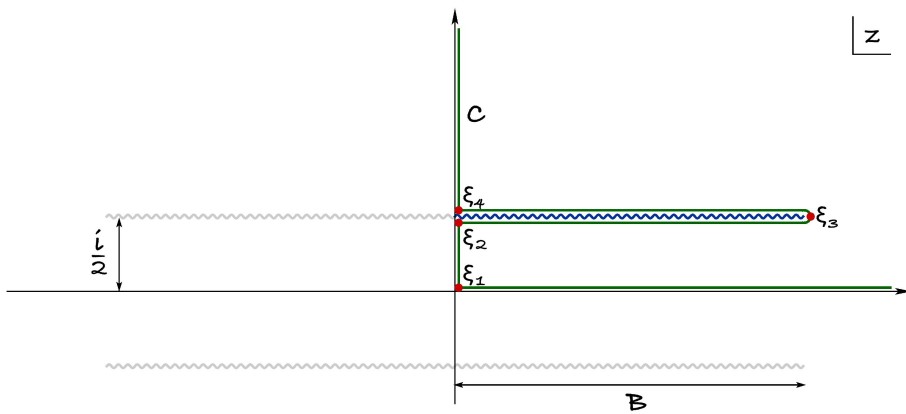

Figure 5: The resolvent is real on the contour $C$ and takes values $\xi_i$ at marked points.

(vi) The explicit form of the resolvent can be found by integrating the following differential equation:

$$dz = g \, \frac{dG(G - \xi_3)}{\sqrt{2(G - \xi_1)(G - \xi_2)(G - \xi_4)}} \,, \tag{203}$$

where $\xi_i$ are the values of $G(z)$ at the points marked red in figure 5. Monotonicity implies that $\xi_1 > \xi_2 > \xi_3 > \xi_4$.

The resolvent can expressed through elliptic integrals but we will never need this explicit expression. The parameters of the solution can be also expressed through elliptic integrals, the $K \equiv K(m)$ and $E \equiv E(m)$:

$$
\begin{aligned}
\xi_1 &= \frac{K[(2-m)K - 2E]}{2\pi^2 g^2} \,, \\
\xi_2 &= \frac{K(K - 2E)}{2\pi^2 g^2} \,, \\
\xi_3 &= \frac{K[(2-m)K - 3E]}{2\pi^2 g^2} \,, \\
\xi_4 &= \frac{K[(1-m)K - 2E]}{2\pi^2 g^2} \,.
\end{aligned}
\tag{204}
$$

The elliptic modulus is implicitly determined by the relation:

$$6\pi^4 g^2 = K^2 \big[ 2(2-m)KE - (1-m)K^2 - 3E^2 \big] \,. \tag{205}$$

While the endpoint of the eigenvalue distribution is given by

$$\pi B = KE(\varphi) - EF(\varphi) \,, \tag{206}$$

where

$$\sin \varphi = \frac{K - E}{mK} \,. \tag{207}$$

A step-by-step derivation of all these results can be found in [22, 23].

The strong-coupling limit corresponds to $m \to 1^-$ is when the elliptic $K$ develops a log-divergence. Denoting

$$L \equiv 4 \log \frac{2}{1 - m} \,, \tag{208}$$

we find:

$$6\pi^4 g^2 \simeq \frac{L^2(L - 3)}{4} \,, \tag{209}$$

and

$$\sin \varphi \simeq \frac{L-2}{L}. \tag{210}$$

Likewise,

$$\xi_1 \simeq \frac{L(L-4)}{8\pi^2 g^2}, \qquad \xi_2 \simeq \frac{L(L-4)}{8\pi^2 g^2}, \qquad \xi_3 \simeq \frac{L(L-6)}{8\pi^2 g^2}, \qquad \xi_4 \simeq -\frac{L}{2\pi^2 g^2}. \tag{211}$$

Corrections are exponential in $L$.

It then follows from (206) that

$$2\pi B \simeq L - \log L - 2, \tag{212}$$

up to corrections of order $1/L$. Solving (209) for $L$ we get, with the same accuracy:

$$B = (3\pi g^2)^{\frac{1}{3}} - \frac{1}{6\pi} \log\left(24\pi^4 g^2 e^3\right). \tag{213}$$

This agrees with (199) in the leading-log approximation, if we recall that $B = \mu/\hbar$ and $\hbar = g^{-2/3}$.

The eigenvalue density can be found from (201) as the discontinuity across the cut in figure 5. We first notice that $\xi_3 \gg \xi_4$ at large $L$, while $\xi_2$ and $\xi_3$ are approximately equal. Since $G(z)$ varies from $\xi_3$ to $\xi_2$ on the lower side of the cut, there it is can be replaced by a constant

$$G\left(a + \frac{i}{2}\right)\bigg|_{\text{lower side}} \simeq \xi_2 \simeq \frac{L^2}{8\pi^2 g^2} \simeq \frac{B^2}{2g^2}, \tag{214}$$

where we have used (211) and (212).

On the upper side, $g^2 G$ varies by a huge amount, from $\mathcal{O}(L)$ to $\mathcal{O}(L^2)$, but both values are big and the asymptotic behaviour (202) can be used to approximate $G(z)$:

$$G\left(a + \frac{i}{2}\right)\bigg|_{\text{upper side}} \simeq \frac{a^2}{2g^2}. \tag{215}$$

Subtracting (215) from the (214) we get, according to (201):

$$\rho(a) \simeq \frac{B^2 - a^2}{4\pi g^2} = \frac{3}{4B^3}(B^2 - a^2). \tag{216}$$

This agrees with the solution for short loops (194). The Wilson loop itself can be found by integrating $e^{xa}$ which makes sense for $x \sim 1/B$. For larger $x$ the exponential is too strong and the integral is dominated by the close vicinity of the endpoint where this crude approximation for the density is no longer valid.

While $G$ itself is very large near the endpoint, $\mathcal{O}(L^2)$, its variation on the scale $\mathcal{O}(L)$ becomes important. It thus becomes necessary to account for the difference between $\xi_2$ and $\xi_3$. At the same time the difference between $\xi_2$ and $\xi_1$ is much smaller and can still be neglected. The differential equation (203) in this approximation becomes

$$dz \simeq \frac{g}{\sqrt{2\xi_3}} dG \frac{G - \xi_3}{G - \xi_2}, \tag{217}$$

and can be integrated in elementary functions. Denoting

$$B + \frac{i}{2} - z \equiv \alpha, \tag{218}$$

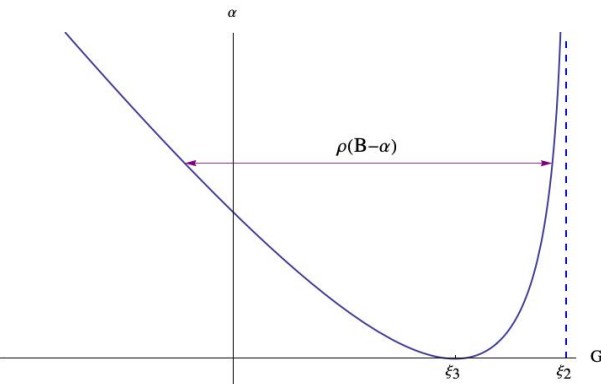

Figure 6: The eigenvalue density is the jump across the branch cut of $G(\alpha)$, equivalently the difference between the two branches for any given $\alpha$.

we find:

$$\alpha = \frac{g}{\sqrt{2\xi_3}}\left[(\xi_2 - \xi_3)\log\frac{\xi_2 - \xi_3}{\xi_2 - G} + \xi_3 - G\right].\tag{219}$$

The integration constant was chosen to ensure $G = \xi_3$ at $\alpha = 0$ (figure 5).

The function $\alpha(G)$ is displayed in figure 6. It is obvious that its inverse $G(\alpha)$ has a branch cut starting at $\alpha = 0$. The density is the discontinuity across the cut and can be found graphically from the definition (201), as shown in the plot. The Wilson loop can be computed as

$$\mathbb{W}(x) = \int_{-B}^{B} da\, \rho(a)\, e^{ax} \simeq e^{Bx}\int_{0}^{\infty} d\alpha\, \rho(B - \alpha)\, e^{-\alpha x}$$

$$= \frac{e^{Bx}}{2\pi}\int_{-\infty}^{\xi_2} dG\,\frac{d\alpha}{dG}\,(G - \xi_3)\, e^{-\alpha x}.\tag{220}$$

Using (217) and (219) this can be explicitly written as

$$\mathbb{W}(x) = \frac{e^{Bx}}{2\pi}\frac{g}{\sqrt{2\xi_3}}\int_{-\infty}^{\xi_2} dG\,\frac{(G - \xi_3)^2}{\xi_2 - G}\, e^{-\frac{gx}{\sqrt{2\xi_3}}\left[(\xi_2 - \xi_3)\log\frac{\xi_2 - \xi_3}{\xi_2 - G} + \xi_3 - G\right]}.\tag{221}$$

Introducing a "dimensionless" variable

$$\xi_2 - G = (\xi_2 - \xi_3)u,\tag{222}$$

and taking into account that

$$\frac{g}{\sqrt{2\xi_3}}(\xi_2 - \xi_3) \simeq \frac{1}{2\pi},\tag{223}$$

we get:

$$\mathbb{W}(x) = \frac{\xi_2 - \xi_3}{4\pi^2}\, e^{(B + \frac{1}{2\pi})x}\int_{0}^{\infty} du\, u^{\frac{x}{2\pi}}(1 - u)^2\, e^{-\frac{ux}{2\pi}}$$

$$= \frac{\xi_2 - \xi_3}{x^2}\,\Gamma\left(1 + \frac{x}{2\pi}\right)e^{x\left(B - \frac{1}{2\pi}\log\frac{x}{2\pi e}\right)}.\tag{224}$$

Using (211), (209) and (213), we finally find:

$$\mathbb{W}(x) = \left( \frac{3}{8\pi^2 g^4 x^6} \right)^{\frac{1}{3}} \Gamma\left(1 + \frac{x}{2\pi}\right) e^{\left[ (3\pi g^2)^{\frac{1}{3}} - \frac{1}{6\pi} \log(3\pi g^2 x^3) \right] x}, \qquad (225)$$

which agrees with the solution of the loop equations for long loops (200). Here we have kept track of the log normalisation and thus computed the constant $C$ in (200) which was left undetermined in the leading-log approximation.

# 7 Conclusion

The multipoint Wilson loop operators in this paper can grant access to connected correlators of matrix fields. Repeating common arguments in literature one can show that our loop equation too is the top one in a tower of equations that interrelate the $n$-point and higher-point operators. A special interest arises in $\mathcal{N} \geqslant 2$ SYM theories: $\mathbb{W}(2\pi k_1, 2\pi k_2, \ldots, 2\pi k_n)$ measures the correlator of $n$ supersymmetric Wilson loops, each wrapping the same circular path $k_i$ times.

For the large part we concentrated on the zeroth, planar order of the topological expansion, where we derived systematic approximations at weak and at strong coupling. Higher-genus results can be generated by the topological recursion, for the models at hand formulated in [14, 15] giving access to potentially arbitrary order in $1/N^2$.

One can venture into finding patterns in perturbative data and engineering closed-form solutions. The task would be rewarding given the scarcity of such formulas, but it may require a dose of ingenuity, judging by the complex nature of the existing results. Simplifications occur in particular models or limits.

We hint at this prospect in section 4.3.2 when our approximate solution was compared to the integral representation of the $M^2$-correction, which is an exact function of $\lambda$. The order $M^4$, which contains products of two zeta numbers, may descend from the double integral of products of Bessel functions. If such step were repeatable for more orders in $M^2$, one would be tell the analyticity properties of the mass series and possibly resum it.

Other interesting questions involve the effect of instantons (to be accounted by an $N$-dependent measure), the large-$N$ and large-$x$ limit with $x\sqrt{\lambda} \sim N$ (with contributions stemming from isolated eigenvalues) for branes in AdS/CFT [58, 85] and the case $x_1 x_2 < 0$ which models Wilson loops in complex-conjugate representations.

Solving the matrix model of circular Wilson loops has been a long-standing dream in $\mathcal{N} = 2$ theories. The loop equation harbours the potential to move the goalpost further into hardly-accessible regions of the parameter space. One can employ finite-difference methods, as noted in section 4.3.2, and the Monte Carlo method for the matrix-model average [41, 72, 75], which is significantly simpler in matrix models in one dimension.

Loops in $\mathcal{N} = 2^*$ theories feature a rich diagram of quantum phase transitions on the line of couplings at infinite mass [34]. The non-commuting limits $\lambda \to 0^-$ and $M \to \infty$ at fixed $\Lambda R$ are markedly different from the perturbation theory in this paper. One could revisit directly in the loop equation and test the findings against the large-$\Lambda R$ asymptotics in [32].

The loop equation is an essential tool to measure observables with a matrix-integral representation. Supersymmetric localisation techniques are available for BPS observables in any $\mathcal{N} = 2$ Lagrangian theories [29]. The prime generalisation is the two circular BPS Wilson loops in $SU(N) \times SU(N)$ quiver theory [39, 46, 83, 84, 86–88]. The model with equal couplings is equivalent to the $\mathbb{Z}_2$ orbifold of $\mathcal{N} = 4$ SYM with gauge group $SU(2N)$ [89] and the loops have the same expectation value: the planar value coincides with that in $\mathcal{N} = 4$ SYM and the non-planar expansion was the subject of a recent study [81]. The model with unequal couplings is the next-to-simplest case where the loop equation could close on and be solvable for

all Wilson loops associated to different gauge groups. Likewise another setup is the class of $A_{q-1}$ circular quiver theories, which are relevant for a number of motivations including integrability in $\mathcal{N} = 2$ theories. Localisation reduces BPS loops to multi-matrix models for which perturbative algorithms are available [86, 90].

The loop equations are applicable to models with fermionic variables, including supereigenvalue models [91–94] or fermionic matrix models [95]. We believe generalisations we discussed are applicable to this class of models as well.

## Acknowledgments

We thank Dmitri Bykov for comments on the $\nu$-model.

**Funding information** The work of E. V. was supported by the European Union's Horizon 2020 research and innovation programme under the Marie Sklodowska-Curie grant agreement No 895958. The work of K. Z. was supported by VR grant 2021-04578. Nordita was partially supported by Nordforsk.

## A Saddle-point equation from loop equation

By the standard argument [8], the saddle-point equations for the eigenvalue integral (4) are equivalent to an integral equations for the eigenvalue density. The latter can be formally defined as a Fourier transform of the Wilson loop (29):

$$\rho(a) = \int\limits_{-\infty}^{+\infty} \frac{d\omega}{2\pi}\, e^{-i\omega a} W(\omega)\,. \tag{A.1}$$

Here we derive the saddle-point equation for the density from the Fourier-space loop equation (30).

The latter can be written as

$$\frac{i}{2} V'\left(-i\frac{\partial}{\partial\kappa}\right) W(\kappa) = \int\limits_{-\infty}^{+\infty} \frac{d\omega}{2\pi}\, \hat{R}(\omega) W(\omega) e^{-\omega\frac{\partial}{\partial\kappa}} W(\kappa)\,, \tag{A.2}$$

where the exponential operator shifts the argument of $W(\kappa)$ to $\kappa - \omega$. Going to the $a$-representation (A.1) and doing the $\omega$-integral with the help of (15) we find:

$$\frac{1}{2} V'\left(-i\frac{\partial}{\partial\kappa}\right) W(\kappa) = \int da\, \rho(a) R\left(-i\frac{\partial}{\partial\kappa} - a\right) W(\kappa)\,. \tag{A.3}$$

Differential operators on both sides act on the same function $W(\kappa)$ and should coincide to give the same result. This condition is equivalent to an integral equation for the density:

$$\frac{1}{2} V'(b) = \int da\, \rho(a) R(b-a)\,, \tag{A.4}$$

where we denoted $b \equiv -i\partial/\partial\kappa$. This is the standard saddle-point equation [8] for the eigenvalue integral (1).

# B  Operator identity and orthogonality relations

In this appendix we derive the operator identity (34) valid for functions whose Fourier transform has a compact support, and derive orthogonality relations for the ensuing differential polynomials.

Substituting (33) in (34) we get:

$$\mathcal{D}_n^{(\nu)} f(\kappa) = -2i \int\limits_0^\infty d\omega\, \omega^{2\nu-2} \frac{J_{\nu+n}(\omega)}{\omega^{\nu+n}} \int\limits_{-1}^1 dt\, e^{it\kappa} \hat{f}(t) \sin t\omega \,. \tag{B.1}$$

Interchanging the order of integration we can evaluate the $\omega$ integral explicitly:

$$\int\limits_0^\infty d\omega\, \omega^{2\nu-2} \frac{J_{\nu+n}(\omega)}{\omega^{\nu+n}} \sin t\omega = \frac{2^{\nu-n-1}\Gamma(\nu)}{\Gamma(n+1)} t\, {}_2F_1\left(\nu,-n;\frac{3}{2};t^2\right)\,. \tag{B.2}$$

It is important to stress that this formula is only valid for $|t| < 1$. For $t$ outside the unit interval the result is way more complicated. The restriction to $|t| < 1$ is sufficient for our purposes since the $t$ integration is confined to the $(-1,1)$ interval by definition, because we only consider functions with a Fourier image of finite support. This is not a technical condition, without it the derivation falls apart and the final conclusion does not hold at all.

The Gamma-function in the denominator hits the pole at negative integer $n$. This immediately implies that the whole integral is to zero for $n < 0$. If $n$ is a non-negative integer, the result is not zero but then the Taylor expansion of the hypergeometric function truncates to a finite polynomial [55]:

$$t\, {}_2F_1\left(\nu,-n;\frac{3}{2};t^2\right) = \frac{(-1)^n \Gamma(\nu-n-1)\Gamma(n+1)}{2\Gamma(\nu)} C_{2n+1}^{\nu-n-1}(t)\,. \tag{B.3}$$

Under the integral $t$ can be replaced by $-i\partial/\partial\kappa$, so any polynomial in $t$ can be replaced by a differential operator acting on $f(\kappa)$. The operator identity (34) then immediately follows, with the operator on the right-hand-side given by the differential polynomial (36).

The polynomials (36) do not satisfy any standard orthogonality relations. That perhaps requires some explanation. The Gegenbauer polynomials of course form an orthonormal set, but that assuming the upper index fixed and the lower one varying. In our case both indices vary with $n$, so polynomials with different $n$ in fact belong to different series with different orthogonality measures.

The closest analogy to classical orthogonality can be derived from the differential equation that the Gegenbauer polynomials satisfy:[32]

$$\left[(1-a^2)\frac{d^2}{da^2} - (2\nu-2n-1)a\frac{d}{da} + (2\nu-1)(2n+1)\right] C_{2n+1}^{(\nu-n-1)}(a) = 0\,. \tag{B.4}$$

The operator in the square brackets is Hermitian on the interval $(-1,1)$ with respect to the measure $(1-a^2)^{\nu-n-\frac{1}{2}}$. We can use this to derive a quadratic integral identity by multiplying both sides of the equation with $C_{2m+1}^{(\nu-m-1)}(a)(1-a^2)^{\nu-n-\frac{1}{2}}$ and integrating by parts:

$$\int\limits_{-1}^1 da(1-a^2)^{\nu-n-\frac{1}{2}} C_{2n+1}^{(\nu-n-1)}(a) \left[(1-a^2)\frac{d^2}{da^2} - (2\nu-2n-1)a\frac{d}{da}\right.$$

$$\left. + (2\nu-1)(2n+1)\right] C_{2m+1}^{(\nu-m-1)}(a) = 0\,. \tag{B.5}$$

---

[32]The conventional orthogonality for a fixed upper index is also a consequence of the same differential equation.

The Gegenbauer polynomial $C_{2m+1}^{(\nu-m-1)}$ satisfies (B.4) with $n$ replaced by $m$. This can be used to simplify the integrand and we finally arrive at an identity

$$2(n-m)\int_{-1}^{1} da\,(1-a^2)^{\nu-n-\frac{1}{2}} C_{2n+1}^{(\nu-n-1)}(a)\left(a\frac{d}{da}+2\nu-1\right)C_{2m+1}^{(\nu-m-1)}(a)=0\,. \qquad (B.6)$$

When $m\neq n$ we can divide by $(n-m)$ and the integral itself must vanish. If $m=n$ we get do not get any useful relation. The $m=n$ integral has to be computed by hand, fortunately is reduces to table integrals, such that

$$\int_{-1}^{1} da(1-a^2)^{\nu-n-\frac{1}{2}} C_{2n+1}^{(\nu-n-1)}(a)\left(a\frac{d}{da}+2\nu-1\right)C_{2m+1}^{(\nu-m-1)}(a)$$

$$=\frac{\pi\Gamma(2\nu-1)}{2^{2\nu-2n-4}(2n+1)!\,\Gamma(\nu-n-1)^2}\,\delta_{nm}\,. \qquad (B.7)$$

Taking into account the normalisation factor in (36), we arrive at the orthogonality condition (48).

# C $U(N)$ vs. $SU(N)$

The $SU(N)$ version of the matrix model constrains the centre of mass of the eigenvalues:

$$Z_{SU(N)}=\int_{-\infty}^{+\infty}\prod_{i=1}^{N} da_i\,\delta\left(\sum_i a_i\right)\prod_{i<j}\mu(a_i-a_j)\,e^{-\frac{1}{2g^2}\sum_i a_i^2}\,, \qquad (C.1)$$

and the same in the correlation functions. In this appendix we consider the model with the Gaussian potential but arbitrary measure. We will prove that the $U(1)$ contribution factors out despite non-linearity of the measure:

$$Z_{U(N)}=Z_{U(1)}Z_{SU(N)}\,, \qquad \mathbb{W}_{U(N)}(x)=\mathbb{W}_{U(1)}(x)\mathbb{W}_{SU(N)}(x)\,, \qquad (C.2)$$

where

$$Z_{U(1)}=\sqrt{2\pi g^2}\,, \qquad \mathbb{W}_{U(1)}(x)=e^{\frac{g^2 x^2}{2}}\,. \qquad (C.3)$$

To do so we Fourier transform the delta function:

$$Z_{SU(N)}=\int\frac{d\omega}{2\pi}\int_{-\infty}^{+\infty}\prod_{i=1}^{N} da_i\prod_{i<j}\mu(a_i-a_j)\,e^{-\frac{1}{2g^2}\sum_i a_i^2+i\omega\sum_i a_i}\,, \qquad (C.4)$$

and shift the integration variables:

$$a_i\rightarrow a_i+ig^2\omega\,. \qquad (C.5)$$

This generates an effective potential for $\omega$: $V(\omega)=g^2\omega^2/2$, and has no effect on the measure since the shift is common to all the eigenvalues. The integral over $\omega$ thus decouples, and we get:

$$Z_{SU(N)}=\frac{1}{\sqrt{2\pi g^2}}Z_{U(N)}\,. \qquad (C.6)$$

The same manipulations over the Wilson loop give:

$$\mathbb{W}_{SU(N)}(x)=\mathbb{W}_{U(N)}(x)\left\langle e^{ig^2 x\omega}\right\rangle=\mathbb{W}_{U(N)}(x)\,e^{-\frac{g^2 x^2}{2}}\,. \qquad (C.7)$$

# D  Derivation of planar equations

We derive (75) and (76). The goal is to insert (69) into (67) and identify the coefficients of $\lambda$ and $x$, ignoring negative powers of $N$. We rescale away $\tilde{c}_{n,\ell,m} \to (16\pi^2)^\ell \tilde{c}_{n,\ell,m}$ for simplicity.

In the left-hand side

$$\sum_{r=2}^{\infty} r\tilde{T}_r \left(\frac{d}{dx}\right)^{r-1} \tilde{\mathbb{W}}(x) = \sum_{r=2}^{\infty}\sum_{\ell=1}^{\infty} \lambda^\ell \sum_{p=r-1}^{2\ell} \frac{r\,p!}{(p-r+1)!} \tilde{T}_r \tilde{c}_{0,\ell,p} x^{p-r+1}, \tag{D.1}$$

we rearrange the sums, shift $p \to p+r-1$

$$\sum_{\ell=1}^{\infty} \lambda^\ell \sum_{r=2}^{\infty}\sum_{p=0}^{2\ell-r+1} \frac{r(p+r-1)!}{p!} \tilde{T}_r \tilde{c}_{0,\ell,p+r-1} x^p, \tag{D.2}$$

and pull out the sum over $x$-powers

$$\sum_{\ell=1}^{\infty} \lambda^\ell \sum_{p=0}^{2\ell-1} x^p \sum_{r=2}^{2\ell-p+1} \frac{r(p+r-1)!}{p!} \tilde{T}_r \tilde{c}_{n,\ell,p+r-1}. \tag{D.3}$$

In the right-hand side

$$\frac{\lambda}{16\pi^2} \int_{-\infty}^{+\infty} \frac{d\omega}{2\pi} \hat{\gamma}(\omega) \int_0^x ds\, \tilde{\mathbb{W}}(s-i\omega)\tilde{\mathbb{W}}(x-s+i\omega), \tag{D.4}$$

we apply the binomial theorem on the powers of $s-i\omega$ and $x-s+i\omega$ and integrate using (73)

$$\frac{1}{16\pi^2} \sum_{n=0}^{\infty}\sum_{n'=0}^{\infty} \lambda^{n+n'+1} \sum_{m=0}^{2n}\sum_{k_1=0}^{m}\sum_{m'=0}^{2n'}\sum_{k_2=0}^{m'}\sum_{k_3=0}^{m'-k_2} (-)^{m'-k_2} \binom{m}{k_1}\binom{m'}{k_2}\binom{m'-k_2}{k_3}$$
$$\times \frac{x^{k_1+k_2+k_3+1}}{k_1+k_3+1} \gamma^{(m+m'-k_1-k_2-k_3)} \tilde{c}_{0,n,m}\tilde{c}_{0,n',m'}. \tag{D.5}$$

A change of indices $\ell = n+n'+1$ brings a homogeneous $\lambda$-power. The exponent of $x$ calls for more dexterity. We rotate $k_2$ and $k_3$ into their sum $s$ and difference $d$

$$\frac{1}{16\pi^2} \sum_{\ell=1}^{\infty} \lambda^\ell \sum_{\ell'=0}^{\ell-1}\sum_{m=0}^{2\ell'}\sum_{m'=0}^{2\ell-2\ell'-2}\sum_{k_1=0}^{m}\sum_{s=0}^{m'} x^{k_1+s+1} \sum_{d=-s,-s+2,\ldots,s} \frac{(-)^{m'-\frac{s+d}{2}}}{k_1+\frac{s-d}{2}+1} \binom{m}{k_1}\binom{m'}{\frac{s+d}{2}}$$
$$\times \binom{m'-\frac{s+d}{2}}{\frac{s-d}{2}} \gamma^{(m+m'-k_1-s)} \tilde{c}_{0,\ell',m}\tilde{c}_{0,\ell-\ell'-1,m'}, \tag{D.6}$$

commute the sums over $\ell'$ and $m,m'$

$$\frac{1}{16\pi^2} \sum_{\ell=1}^{\infty} \lambda^\ell \sum_{m=0}^{2\ell-2}\sum_{m'=0}^{2\ell-2\lceil\frac{m}{2}\rceil-2}\sum_{\ell'=\lceil\frac{m}{2}\rceil}^{\ell-\lceil\frac{m'}{2}\rceil-1}\sum_{k_1=0}^{m}\sum_{s=0}^{m'} x^{k_1+s+1}$$
$$\times \sum_{d=-s,-s+2,\ldots,s} \frac{(-)^{m'-\frac{s+d}{2}}}{k_1+\frac{s-d}{2}+1} \binom{m}{k_1}\binom{m'}{\frac{s+d}{2}}\binom{m'-\frac{s+d}{2}}{\frac{s-d}{2}} \gamma^{(m+m'-k_1-s)} \tilde{c}_{0,\ell',m}\tilde{c}_{0,\ell-\ell'-1,m'}, \tag{D.7}$$

pull out the sums over the indices of $x$

$$
\frac{1}{16\pi^2} \sum_{\ell=1}^{\infty} \lambda^\ell \sum_{k_1=0}^{2\ell-2} \sum_{s=0}^{2\ell-2} x^{k_1+s+1} \sum_{m=k_1}^{2\ell-2} \sum_{m'=s}^{2\ell-2} \sum_{\ell'=\lceil \frac{m}{2} \rceil}^{\ell-\lceil \frac{m'}{2} \rceil-1} \sum_{d=-s,-s+2,\ldots,s} \frac{(-)^{m'-\frac{s+d}{2}}}{k_1+\frac{s-d}{2}+1} \binom{m}{k_1} \binom{m'}{\frac{s+d}{2}}
$$
$$
\times \binom{m'-\frac{s+d}{2}}{\frac{s-d}{2}} \gamma^{(m+m'-k_1-s)} \tilde{c}_{0,\ell',m} \tilde{c}_{0,\ell-\ell'-1,m'}, \tag{D.8}
$$

and rotate $k_1, s$ into $p, q$ to create $x^p$

$$
\frac{1}{16\pi^2} \sum_{\ell=1}^{\infty} \lambda^\ell \sum_{p=1}^{2\ell-1} x^p \sum_{q=1-p,3-p,\ldots,p-1} \sum_{m=\frac{p-q-1}{2}}^{2\ell-2} \sum_{m'=\frac{p+q-1}{2}}^{2\ell-2} \sum_{\ell'=\lceil \frac{m}{2} \rceil}^{\ell-\lceil \frac{m'}{2} \rceil-1}
$$
$$
\times \sum_{d=-\frac{p+q-1}{2},-\frac{p+q-5}{2},\ldots,\frac{p+q-1}{2}} \binom{m}{\frac{p-q-1}{2}} \binom{m'}{\frac{p+q-1}{4}+\frac{d}{2}} \binom{m'-\frac{p+q-1}{4}-\frac{d}{2}}{\frac{p+q-1}{4}-\frac{d}{2}} \frac{(-)^{m'-\frac{p+q-1}{4}-\frac{d}{2}}}{\frac{3p-q+1}{4}-\frac{d}{2}}
$$
$$
\times \gamma^{(m+m'-p+1)} \tilde{c}_{0,\ell',m} \tilde{c}_{0,\ell-\ell'-1,m'}. \tag{D.9}
$$

We equate the powers of $\lambda$ and $x$ across (D.3) and (D.9), undo $\tilde{c}_{n,\ell,m} \to \tilde{c}_{n,\ell,m}/(16\pi^2)^\ell$ and prove (75) and (76).

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
