# Peer review of "Loop equations for generalised eigenvalue models"

_SciPost Physics, doi:SciPost Phys. 17, 017 (2024)_

## Round 2 · Referee Report · Anonymous (Referee 1) · 2024-5-10

Report

I know this paper and I think that it deserves publication in SciPost in its present form.

It concerns a relatively novel subject of eigenvalue matrix models with non-trivial (non-Vandermonde) measures, which are not openly integrable -- and approaches it from the side of loop equations.
No striking results are obtained yet -- but this is natural at this stage of investigation, when different attempts are made and intuition about the possible hidden properties is slowly developed.

Recommendation

Publish (meets expectations and criteria for this Journal)

---

## Round 2 · Referee Report · Anonymous (Referee 2) · 2024-6-2

Report

This paper treats a the problem of evaluating matrix integrals with non-Vandermond measure which appear in the computation of some observables in supersymmetric SU(N)-invariant Yang-Mills theories. The SU(N) symmetry reduces the problem to the statistical system of N particles on al line with repulsive interaction. The system is solvable only if there is an extra infinite symmetry as in the case of beta-ensembles. The authors derive loop equations for general interaction potential. The loop equations have the same structure as the standard ones, but the non-Vandermond part of the measure produces an extra weight for the contact term which is not a convolution anymore. Then the authors explore the advantages and the limits of the method based on these generalised loop equations. This is a valuable paper and I recommend publication of the manuscript in its present form.

Recommendation

Publish (easily meets expectations and criteria for this Journal; among top 50%)

---

## Round 2 · Referee Report · Anonymous (Referee 3) · 2024-6-10

Strengths

1) It addresses a more general type of matrix models, which are not very much studied in spite of appearing often in applications.

2) It contains many different examples and applies its ideas and techniques to various models.

Weaknesses

Connection to solutions based on topological recursion is not developed in detail.

Report

Generalised eigenvalue models are multidimensional integrals which generalize the eigenvalue representation of conventional matrix models. Typically they have an interaction between eigenvalues which generalizes the usual Vandermonde determinant. These models appear in many contexts, like Chern-Simons theory, localization of supersymmetric gauge theories, etc. The solution of these models in the 1/N expansion is still insufficiently studied, and this paper intends to study these models by using the well-known method of loop equations. The paper develops this technique and then looks for solutions in the weak coupling regime, as a power series expansion in the coupling constants (and 1/N). It also studies the strong coupling regime -a new approach to these models which is particularly useful when there is a string theory dual.

The strength of this paper is its down to earth approach, and the many examples that are discussed and worked out in detail. On the other hand, the loop equations of the conventional matrix model can be solved explicitly by topological recursion, and this paper does not provide an extension thereof for the generalized eigenvalue models. Moreover, the very same type of models were studied in detail in the topological recursion literature, in papers by Borot, Eynard and Orantin (1303.5808) and by Borot (1307.4957). These papers worked out their loop equations, as in the paper under referee, and presented a solution to them in terms of a generalization of topological recursion. I think that the current paper should not only cite these papers, but compare their results in some detail, when possible.

In conclusion, I would recommend publication of a revised version of the paper in which the relation to the two papers mentioned above is clarified in some detail.

Requested changes

Discuss in some detail the relation between the results in this paper and the solution of the same models via topological recursion due to Borot- Eynard-Orantin and Borot.

Recommendation

Ask for minor revision

---

## Round 3 · Author Response

We would like to thank the referee for bringing our attention to 1303.5808, 1307.4957
highly relevant for our work. These results and ours, we believe, are complementary as we mostly discuss genus-0
expectation values. Those are treated as an input in 1303.5808, 1307.4957 to generate
the whole topological expansion (if we misinterpret these results we are open for further suggestions).
We added a paragraph in the introduction on the relation of our work to 1303.5808, 1307.4957, and also comment on it in the beginning
of the conclusions.
highly relevant for our work. These results and ours, we believe, are complementary as we mostly discuss genus-0
expectation values. Those are treated as an input in 1303.5808, 1307.4957 to generate
the whole topological expansion (if we misinterpret these results we are open for further suggestions).
We added a paragraph in the introduction on the relation of our work to 1303.5808, 1307.4957, and also comment on it in the beginning
of the conclusions.

---

## Round 3 · List of Changes

- Added refs [13] and [14]
- Added the fourth paragraph on p. 2 and the second paragraph in sec. 7 discussing the relation between this work and our results

---

## Editorial Decision

published